# Unified Language-Vision Pretraining in LLM with Dynamic Discrete Visual Tokenization

**Yang Jin**[1]*, **Kun Xu**[2], **Kun Xu**[2], **Liwei Chen**[2], **Chao Liao**[2], **Jianchao Tan**[2],
**Quzhe Huang**[1], **Bin Chen**[2], **Chenyi Lei**[2], **An Liu**[2], **Chengru Song**[2],
**Xiaoqiang Lei**[2], **Di Zhang**[2], **Wenwu Ou**[2], **Kun Gai**[2], **Yadong Mu**[1]†

[1]Peking University    [2]Kuaishou Technology

jiny@stu.pku.edu.cn, {xukunxkxk,syxu828}@gmail.com, myd@pku.edu.cn

## Abstract

Recently, the remarkable advance of the Large Language Model (LLM) has inspired researchers to transfer its extraordinary reasoning capability to both vision and language data. However, the prevailing approaches primarily regard the visual input as a prompt and focus exclusively on optimizing the text generation process conditioned upon vision content by a frozen LLM. Such an inequitable treatment of vision and language heavily constrains the model's potential. In this paper, we break through this limitation by representing both vision and language in a unified form. Specifically, we introduce a well-designed visual tokenizer to translate the non-linguistic image into a sequence of discrete tokens like a foreign language that LLM can read. The resulting visual tokens encompass high-level semantics worthy of a word and also support dynamic sequence length varying from the image. Coped with this tokenizer, the presented foundation model called **LaVIT** can handle both image and text indiscriminately under the same generative learning paradigm. This unification empowers LaVIT to serve as an impressive generalist interface to understand and generate multi-modal content simultaneously. Extensive experiments further showcase that it outperforms the existing models by a large margin on massive vision-language tasks. Our code and models are available at https://github.com/jy0205/LaVIT.

## 1 Introduction

The large language models (LLMs) (Brown et al., 2020; Touvron et al., 2023) nowadays have demonstrated impressive advances in various linguistic applications. Profiting from the knowledge in the massive text corpus, they possess exceptional understanding capabilities and serve as a general-purpose interface to complete a wide range of real-world tasks. Such success has motivated researchers to investigate the Multi-modal Large Language Models (MLLMs), which aim at extending the powerful pure-text LLMs to process multi-modality inputs. As shown in Figure 1-(a), the prevalent approaches mostly leverage an adapter architecture (e.g., the Resampler (Alayrac et al., 2022), linear projection (Liu et al., 2023), or Q-Former (Li et al., 2023)) to map the visual features encoded by a pre-trained vision backbone (Radford et al., 2021) to the semantic space of LLM.

Despite achieving preliminary results in zero-shot multi-modal understanding, they still suffer from inherent design deficiencies. The training objective of prior methodologies (Li et al., 2023; Huang et al., 2023; Zhu et al., 2023) is centered on predicting textual descriptions dependent on visual content, where the visual parts are merely regarded as prompts without any supervision. The inequitable treatment of different modal inputs severely constrains the model's potential, limiting them to only performing comprehension tasks like generating text based on images. Moreover, most of these methods completely delegate the responsibility of vision-language alignment to the newly added adapter with limited trainable parameters, which fails to leverage the remarkable reasoning capabilities of LLM to learn the interaction across different modalities. Although the recent concurrent work

---

*Work done during an internship at Kuaishou Technology.
†Corresponding Author.

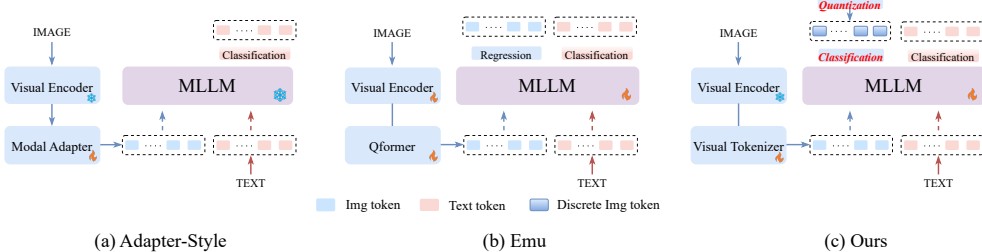

Figure 1: The comparisons between different MLLMs. (a) The adapter-style methods rely on an adapter network to project visual features into the semantic space of LLM. During training, visual tokens are merely treated as the prompt to guide text generation. (b) The concurrent work Emu adopts the regression loss for visual features and jointly trains with textual tokens. (c) We craft a visual tokenizer to represent images in the same discrete format as text so as to indiscriminately optimize them under a unified generative objective.

Emu (Sun et al., 2023) proposes to unlock the text-pretrained LLM by regressing the next visual embedding during pre-training (Figure 1-(b)), the inconsistent optimization objectives for image and text are not conducive to unified multi-modal modeling.

In this work, we introduce **LaVIT** (**La**nguage-**VI**sion **T**ransformer), a novel general-purpose multi-modal foundation model that inherits the successful learning paradigm of LLM: predicting the next image/text token in an auto-regressive manner. Our insight is that by employing a unified objective to indiscriminately treat tokens from different modalities, the model can seamlessly achieve "any-to-any" multi-modal comprehension and generation. However, the original LLM is specifically crafted to process discrete textual tokens. When dealing with physical signal inputs, such as images, it becomes imperative to embrace a representation seamlessly compatible with text tokens. Therefore, we propose to translate the image into a sequence of tokens like a foreign language that LLM can comprehend, so that both images and texts can be handled simultaneously under the unified generative objective without any specific architectural modification, as shown in Figure 1-(c).

To achieve this goal, a crucial element lies in the development of an efficient visual tokenizer for encoding images, which we contend should adhere to the following principles: (i) **discrete visual token**: While language models rely on text tokens defined by a dictionary, prior visual tokens, like those derived from ViT, consist of continuous feature vectors encoding a patch. In approaches such as masked image modeling (He et al., 2022) or masked feature prediction (Wei et al., 2022), regressive objectives on continuous features or raw visual pixels are employed for self-supervised pretraining. Here, we advocate for quantizing the visual tokens into a discrete form, aligning them with the next-token prediction objective in language models. This form is particularly advantageous when the target distribution for the next token is multi-mode. (ii) **dynamic token allocation**. Given the varying semantic complexity of different images, employing a fixed length of tokens to encode all images is compute-uneconomical. Moreover, as a key difference from textual tokens, visual patches exhibit a notable interdependence, making it considerably more straightforward to deduce one token from others. This renders the next-token paradigm less effective in learning visual knowledge through self-supervision. Thus we argue for the token-merging to ensure the least redundancy among visual patches, thereby rendering a dynamic token number for different images.

Following the aforementioned two crucial fundamentals, LaVIT introduces a novel dynamic visual tokenization mechanism consisting of a selector and a merger to process images. The token selector first decides which visual patches carry informative semantics and are necessary to be selected to encode the whole image. In order to maximally preserve the image details, the token merger further compresses the unselected patches onto the retained ones according to their feature similarity. Such a design enables each retained visual token to contain high-level semantics from multiple similar patches and thus reduce the redundancy among tokens. This selecting and merging strategy will produce a dynamic sequence length varying from the image content itself. The retained visual tokens are further quantized into discrete codes by a learnable codebook (Esser et al., 2021), which will serve as the supervision signals for visual tokens during pre-training. Empowered by this visual tokenizer, our LaVIT can be trained with a simple yet unified objective: predicting the next image/text token in the multi-modal sequence. After pre-training, LaVIT can serve as a multi-modal generalist to perform both multi-modal comprehension and generation without further fine-tuning (See Figure 2). The key contributions of this work are summarized as:

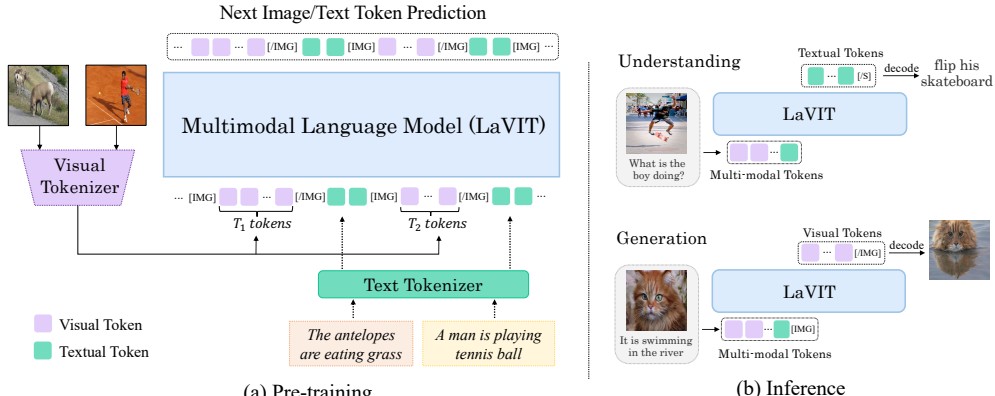

Figure 2: Given an image-text pair, the image is tokenized into discrete tokens and concatenated with text tokens to form a multi-modal sequence. Then, LaVIT is optimized under a unified generative objective. After training, it can achieve both zero-shot multi-modal comprehension and generation.

- We introduce LaVIT, a new effective, general-purpose multi-modal foundation model that goes beyond the traditional adapter-based architectures. By transforming images into a sequence of discrete tokens like a foreign language that LLM can comprehend and generate, both modalities can be associated indiscriminately under a unified generative training paradigm.

- The developed visual tokenizer can produce discrete visual tokens with dynamic length to reduce the interdependence among visual patches, which enhances the representation compatibility of image and text in LLM and improves computational efficiency.

- Our LaVIT showcases the extraordinary multi-modal understanding and generation potential. It can take any modality combinations as input and perform impressive in-context generation of both images and text. As demonstrated by extensive experiments, LaVIT achieves state-of-the-art zero-shot performance on a wide range of vision-language tasks.

## 2 RELATED WORK

**Vision-Language Pre-training.** Researchers have extensively investigated vision-language pre-training (VLP). The pioneer works (Radford et al., 2021; Jia et al., 2021) primarily employ dual-encoder with contrastive objectives to learn the generic cross-modal aligned representations. Recently, the rapid progress of large language models (Chowdhery et al., 2022; Touvron et al., 2023) has motivated researchers to delve into the exploration of augmenting LLM towards vision language tasks. The majority of these works adopt an adapter-style network (Zhang et al., 2023) that serves as an intermediate bridge connecting the pre-trained vision encoder and frozen language model. For instance, Flamingo (Alayrac et al., 2022) develops a Perceiver Resampler to generate text-aligned visual representations. Follow-up methods (Li et al., 2023; Zhu et al., 2023) mainly adopt the Q-Former to project the visual semantics to the LLM's input space. However, visual inputs in these methods (Huang et al., 2023; Alayrac et al., 2022) are only considered as the prompt and not involved in the optimization, which heavily restricts the model potential.

**Vector Quantization in Computer Vision.** Vector quantization (Gray, 1984; Nasrabadi & King, 1988) is widely used in image-generative models. The VQ-VAE (Van Den Oord et al., 2017) and DALL-E (Ramesh et al., 2021) proposed to convert an image into a set of discrete codes in a learnable discrete latent space by learning to reconstruct the original image pixels. Models like VQ-GAN (Esser et al., 2021) and ViT-VQGAN (Yu et al., 2021) leverage adversarial and perceptual objectives to further enhance the image generation quality. The BEIT series of works also adopts the quantized visual codes as the supervision in mask image modeling (Peng et al., 2022; Wang et al., 2023). However, most of these methods tokenize the image into a token sequence with a fixed length (e.g., 512 or 1024). Such a long sequence will invariably result in an excessive computational burden. On the contrary, our proposed visual tokenizer reduces the redundancy among image patches and supports dynamic token length, thus enabling efficient multi-modal inference.

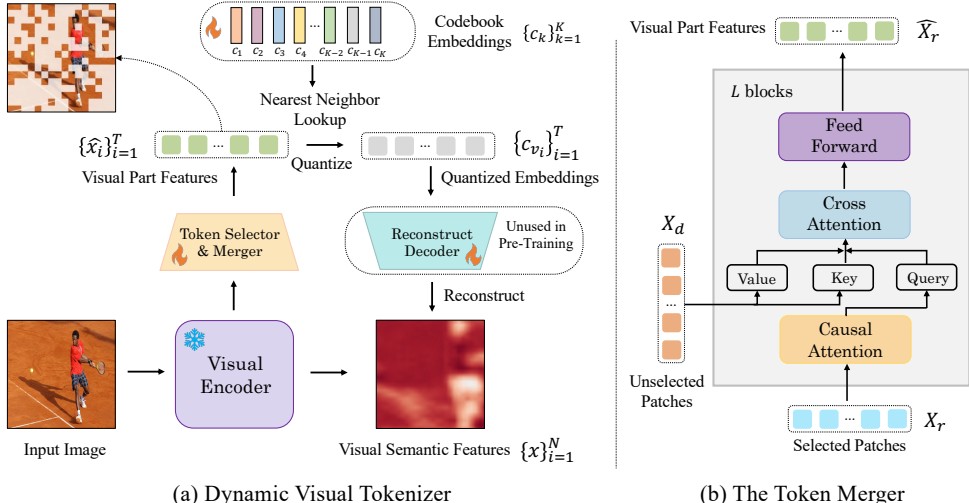

(a) Dynamic Visual Tokenizer

(b) The Token Merger

Figure 3: (a) The pipeline of the proposed dynamic visual tokenizer. It employs a token selector to select the most informative patches and a token merger to compress the information of discarded patches onto the retained ones. The whole tokenizer is trained by maximally reconstructing the semantics of the input image. (b) The detailed architecture of token merger.

## 3 METHOD

This work proposes to leverage the extraordinary reasoning potential of the large language model to facilitate the modeling of both vision and language modalities. In pursuit of this goal, the key component is to represent these two modalities in a uniform form, so as to exploit LLM's successful learning recipe (i.e., next-token prediction). As shown in Figure 2, we develop a visual tokenizer (Section 3.1) to convert the non-linguistic image to the input that LLMs can comprehend. It receives the vision features from a pre-trained vision encoder and outputs a sequence of discrete visual tokens possessing word-like high-level semantics. Coped with the crafted tokenizer, the visual input can be integrated with textual tokens to compose a multi-modal sequence, which is subsequently fed into large language models under a unified auto-regressive training objective (Section 3.2).

### 3.1 STAGE-1: DYNAMIC VISUAL TOKENIZER

Given an image $x \in \mathcal{R}^{H \times W \times C}$, it is first partitioned into $N = HW/P^2$ non-overlapping patches, where $P$ is the patch size. These patches are fed into a pre-trained ViT encoder (Fang et al., 2023) to produce a sequence of the patch features $X = \{x_1, ..., x_N\}$. Then, a straightforward way to encode images is to directly quantize the $N$ patch-level embeddings into discrete tokens as the input to LLMs. This will result in a long visual sequence and bring superfluous computational burden since many visual patches may contain repetitive and trivial background semantics. These redundant patches demonstrate a discernible interdependence, thereby diminishing the efficacy of the next-token paradigm in learning visual knowledge via self-supervision. Consequently, the proposed tokenizer aims to produce visual tokens with a dynamic length according to the complexity of the image content itself. As illustrated in Figure 3, it comprises a token selector and a token merger.

**Token Selector** The token selector takes the $N$ patch-level features $X$ as input. It aims to estimate the importance of each image patch and selects the most informative ones that are competent enough to represent the semantics of the whole image. Inspired by (Rao et al., 2021), it is implemented as a lightweight module consisting of several MLP layers to predict a distribution $\pi \in \mathcal{R}^{N \times 2}$, where $\pi_i = \text{MLP}(x_i)$. By sampling from the distribution $\pi$, a binary decision mask $M \in \{0, 1\}^N$ can be generated, which indicates whether to remain the corresponding image patch. To relax the sampling to be differentiable, the Gumbel-Softmax trick (Maddison et al., 2016) is applied to $\pi$:

$$\hat{\pi_{i,j}} = \frac{\exp((\log \pi_{i,j} + G_{i,j})/\tau)}{\sum_{r=1}^{2} \exp((\log \pi_{i,r} + G_{i,r})/\tau)}. \tag{1}$$

where $G_{i,j}$ is the noise sampled from a Gumbel distribution, $\tau$ is the temperature to control smoothness. Then, the binary decision mask $M$ can be sampled from $\hat{\pi}$ for end-to-end training.

**Token Merger** According to the generated decision mask, total $N$ image patches can be partitioned into retained and dropped groups, with $T$ and $N - T$ patches respectively, denoted as $X_r = \{x_i\}_{i=1}^T$ and $X_d = \{x_j\}_{j=1}^{N-T}$. Instead of directly discarding $X_d$, we develop a token merger to deal with it to maximally preserve the detailed semantics of the input image. As shown in the right of Figure 3, the token merger will progressively compress the information of discarded $X_d$ onto the retained $X_r$ according to their semantic similarity. Concretely, it consists of $L$ stacked blocks, each of which has a causal self-attention layer, a cross-attention layer, and a feed-forward layer. In the causal self-attention layer, each token in $X_r$ attends to its previous tokens with a causal mask. This helps to convert 2D raster-ordered features from the ViT encoder into a sequence with causal dependency, thus ensuring consistency with textual tokens in LLMs. We found this strategy can result in better performance than bi-directional self-attention. The cross-attention layer treats the retained tokens $X_r$ as the query and merges tokens in $X_d$ based on their similarity in the embedding space. Formally, this layer calculates an update of $X_r$ by:

$$\Delta X_r = \text{softmax}\left(QK^\top/\sqrt{D}\right) V, \tag{2}$$

where $D$ denotes the dimension of hidden state, $Q = W_Q X_r \in \mathcal{R}^{T \times D}$, $K = W_K X_d \in \mathcal{R}^{(N-T) \times D}$ and $V = W_V X_d \in \mathcal{R}^{(N-T) \times D}$. To parallelize the computation, we adopt the predicted decision mask $M$ to control the cross-attention scope between tokens without directly partitioning them into two groups. After $L$ successive token merger blocks, we can obtain the final merged visual tokens $\hat{X}_r = \{\hat{x}_i\}_{i=1}^T$. Each token implicitly encodes high-level semantics from several image patches possessing similar visual patterns, which we refer to as visual part features $\hat{X}_r$. The token selector and merger work together to dynamically adjust the visual token sequence length to accommodate images with different content complexity.

**Vector Quantization and Training** The generated visual part features $\hat{X}_r$ are then passed into a quantizer. It tokenizes $\hat{X}_r$ to a sequence of discrete visual codes $V = \{v_i\}_{i=1}^T$ by looking up a learnable codebook $\mathcal{C} = \{c_k\}_{k=1}^K$, where $K$ is codebook size. To be specific, the $i_{th}$ visual code is calculated by assigning $\hat{x}_i$ in $\hat{X}_r$ to its closest neighbourhood code in $\mathcal{C}$:

$$v_i = \arg\min_j \|l_2(\hat{x}_i) - l_2(c_j)\|_2, \quad v_i \in [0, K-1], \tag{3}$$

where $l_2$ indicates the $L_2$ norm. Based on the indexing visual codes, we can obtain the quantized embeddings $\{c_{v_i}\}_{i=1}^T$, which is fed into a decoder to reconstruct the original visual semantic features $X = \{x_i\}_{i=1}^N$. The insight behind this design is that the reconstruction quality of the image semantics depends on selecting the most informative patches (token selector), along with maximally preserving the visual details only through the remained tokens (token merger). Thus, both token selector and merger can be effectively updated by encouraging a higher semantic reconstruction quality. The final training objective of the visual tokenizer is defined as:

$$\mathcal{L}_{\text{tokenizer}} = \frac{1}{N} \sum_{i=1}^N \left(1 - \cos(x_i, x_i^{\text{rec}})\right) + \lambda(\rho - \frac{1}{N} \sum_{i=1}^N M_i)^2, \tag{4}$$

where $\cos(x_i, x_i^{\text{rec}})$ calculates the cosine similarity between the reconstructed and real visual embeddings, $\rho$ is a pre-defined rate that controls the target mean percentage of the retained visual tokens and $\lambda$ is set to be 2. Finally, the tokenized discrete codes $\{v_i\}_{i=1}^T$ will serve as the supervision signals for visual tokens in the following pre-training.

**Decoding to Pixels** The proposed visual tokenizer is capable of reconstructing visual features of input images that contain high-level semantics to represent the image content but lose the pixel-level details. To recover the original pixel space, we employ a conditional de-noising U-Net $\epsilon_\theta$ (Rombach et al., 2022) to infill the visual details after training the visual tokenizer. Specifically, it takes the reconstructed $x_{\text{rec}}$ as the condition to progressively recover image $x$ from a Gaussian noise. Following Rombach et al. (2022), the parameters $\theta$ of this U-Net are optimized by $\epsilon$ prediction:

$$\mathcal{L}_\theta = \mathbb{E}_{z,\epsilon \sim \mathcal{N}(0,1),t} \left[\|\epsilon - \epsilon_\theta(z_t, t, x_{\text{rec}})\|\right], \tag{5}$$

where $z_t$ is the latent state of image $x$ in the diffusion process. We present some pixel decoding results by the trained denoising U-Net in Figure 7 of the appendix. During inference, the generated visual tokens from LaVIT can be decoded into realistic images by this U-Net.

## 3.2 STAGE-2: UNIFIED GENERATIVE MODELING

Given an image-text pair, the 2D image can be tokenized into a 1D sequence with causal dependency and then concatenated with text tokens to form a multi-modal sequence $y = (y_1, y_2, .., y_S)$. For differentiating between two modalities, two special tokens [IMG] and [/IMG] are inserted into the beginning and end of the image token sequence respectively, indicating the start and end of image content. To empower LaVIT with the capability to generate both text and images, we employ two different concatenation forms, i.e., [image, text] and [text; image]. When the image is used as a condition (on the left) to generate text, we use the continuous visual features $\hat{X}_r$ from the token merger instead of quantized visual embeddings as the input to LLMs. Such a design mitigates the loss of detailed information caused by vector quantization, which is crucial for fine-grained multi-modal understanding tasks like visual question answering. Our **LaVIT** adopts the general Language Modeling (LM) objective to directly maximize the likelihood of each multi-modal sequence in an auto-regressive manner:

$$p(y) = \sum_{y \in \mathcal{D}} \sum_{i=1}^{S} \log P_\theta(y_i | y_{<i}). \tag{6}$$

Since both image and text are already represented as discrete token IDs, we can use the cross-entropy to supervise the token prediction at each location for both two modalities with a shared prediction head. The complete unification in representation spaces and the training paradigms can help LLMs better learn multi-modal interaction and alignment. When LaVIT is pre-trained, it possesses the capacity to perceive images akin to a foreign language, comprehending and producing them like text. Nevertheless, most of the existing works merely regard images as prompts to guide the generation of text with no supervision, restricting them to solely performing image-to-text tasks.

## 3.3 MODEL PRE-TRAINING

The LaVIT undergoes a two-stage pre-training procedure on web-scale multi-modal corpora.

**Stage-1: Tokenizer Training**. Following the existing MLLMs, the ViT-G/14 of EVA-CLIP (Fang et al., 2023) is employed as the visual encoder. The visual codebook size is empirically set to $K = 16384$. We adopt $L = 12$ transformer blocks for both token merger and decoder in our tokenizer. During training, this encoder is kept frozen and only the parameters of the selector, merger, and codebook are updated. It is trained for 50K steps on about 100M images from LAION-400M (Schuhmann et al., 2021) with the batch size of $2048$ and $\rho = 1/3$. After training the tokenizer, the conditional U-Net for pixel decoding is initialized from the Stable Diffusion v1.5 (Rombach et al., 2022) and finetuned 20k steps on the same dataset. The whole stage-1 training only requires pure image data without corresponding captions.

**Stage-2: Unified Vision-Language Pre-training**. Based on the trained visual tokenizer, all the images can be tokenized into discrete codes that are amenable to the next token prediction. We utilize the raw 7B version of LLaMA (Touvron et al., 2023) as the default LLM. For image-to-text comprehension (i.e., [image, text]), we employ about 93M samples from Conceptual Caption (Sharma et al., 2018; Changpinyo et al., 2021), SBU (Ordonez et al., 2011), and BLIP-Capfilt (Li et al., 2022). For the text-to-image synthesis (i.e., [text, image]), an additional 100M image-text pairs from the LAION-Aesthetics (A high-aesthetics image subset of LAION-5B (Schuhmann et al., 2022)) are used following Stable Diffusion. Moreover, to reduce catastrophic forgetting of the reasoning capacity in training LLM, we employ the English text corpus from Redpajama (Computer, 2023) dataset and mix it with the above image-text pairs to form the multi-modal input sequence.

## 4 EXPERIMENTS

In this section, comprehensive experiments are conducted to systematically validate the effectiveness of LaVIT on a wide range of vision-language tasks. Specifically, we mainly evaluate the model's zero-shot multi-modal understanding and generation capacity.

### 4.1 ZERO-SHOT MULTIMODAL UNDERSTANDING

We first quantitatively evaluate the zero-shot multi-modal understanding capacity of LaVIT on Image Captioning (NoCaps (Agrawal et al., 2019), Flickr30k (Plummer et al., 2015))and Visual

| Method | Image Captioning | | Visual Question Answering | | | |
|---|---|---|---|---|---|---|
| | Nocaps | Flickr | VQAv2 | OKVQA | GQA | VizWiz |
| Flamingo-3B (Alayrac et al., 2022) | - | 60.6 | 49.2 | 41.2 | - | 28.9 |
| Flamingo-9B (Alayrac et al., 2022) | - | 61.5 | 51.8 | 44.7 | - | 28.8 |
| OpenFlamingo-9B (Awadalla et al., 2023) | - | 59.5 | 52.7 | 37.8 | - | 27.5 |
| MetaLM (Hao et al., 2022) | - | 43.4 | 41.1 | 11.4 | - | - |
| Kosmos-1 (Huang et al., 2023) | - | 67.1 | 51.0 | - | - | 29.2 |
| Kosmos-2 (Peng et al., 2023) | - | 80.5 | 51.1 | - | - | - |
| BLIP-2 (Vicuna-7B) (Li et al., 2023) | 107.5 | 74.9 | - | - | 41.3 | 25.3 |
| BLIP-2 (Vicuna-13B) (Li et al., 2023) | 103.9 | 71.6 | - | - | 32.3 | 19.6 |
| CM3Leon-7B (Yu et al., 2023) | - | - | 47.6 | - | - | 37.6 |
| Emu (LLaMA-13B) (Sun et al., 2023) | - | - | 52.0 | 38.2 | - | 34.2 |
| Ours (LLaMA-7B) | **114.2** | **83.0** | **66.0** | **54.6** | **46.8** | **38.5** |

Table 1: Overview of zero-shot evaluation on multi-modal understanding tasks. Compared with previous methods, our LaVIT achieved the best performance on both benchmarks.

Question Answering (VQAv2 (Goyal et al., 2017), OKVQA (Marino et al., 2019), GQA (Hudson & Manning, 2019), VizWiz (Gurari et al., 2018)). For visual question answering tasks, we use a simple prompt: "Question: {} Answer: {}". The widely-used CIDEr score and VQA accuracy are employed as metrics to evaluate captioning and question answering, respectively.

The detailed performance comparisons are shown in Table 1. As observed, LaVIT surpasses all the existing MLLMs by a large margin on these understanding tasks. For example, it achieves a CIDEr score of 83.0 on the Flickr30k test dataset, compared to 61.5 and 74.9 for the Flamingo-9B and BLIP-2 (Vicuna-7B) under the same scale of model size, respectively. The performance superiority on OKVQA (54.6% v.s. 44.7% of Flamingo-9B) further showcases the excellent multi-modal understanding capacity of LaVIT, since this benchmark contains questions requiring commonsense knowledge and reasoning about the content of images.

| Method | Model Type | FID(↓) |
|---|---|---|
| *Text2Image Specialist:* | | |
| DALL-E (Ramesh et al., 2021) | Autoregressive | 28.0 |
| CogView (Ding et al., 2021) | Autoregressive | 27.1 |
| SD (Rombach et al., 2022) | Diffusion | 12.6 |
| GLIDE (Nichol et al., 2021) | Diffusion | 12.2 |
| DALL-E2 (Ramesh et al., 2022) | Diffusion | 10.4 |
| Make-A-Scene (Gafni et al., 2022) | Autoregressive | 11.8 |
| MUSE-7.6B (Chang et al., 2023) | Non-Autoregressive | 7.9 |
| Imagen-3.4B (Saharia et al., 2022) | Diffusion | 7.3 |
| Parti-20B (Yu et al., 2022) | Autoregressive | **7.2** |
| *Multimodal Large Langauge Model:* | | |
| GILL (OPT-6.7B) (Koh et al., 2023) | LLM | 12.2 |
| Emu (LLaMA-13B) (Sun et al., 2023) | LLM | 11.7 |
| CM3Leon-7B (Yu et al., 2023) | LLM | 10.8 |
| Ours (LLaMA-7B) | LLM | **7.4** |

Table 2: The zero-shot text-to-image generation performance of different models on MS-COCO-30K evaluation benchmark.

It is worth noting that, although the concurrent method Emu (Sun et al., 2023) also leverages the LLM to jointly model the vision and language, the direct feature regression objective for visual inputs makes it incompatible with text input. Therefore, despite using more training data (2.6B image-text pairs and 3.8B web-crawled data) and larger LLM (LLaMA 13B), it still achieves inferior performance to ours on all evaluation benchmarks.

## 4.2 ZERO-SHOT MULTIMODAL GENERATION

Since the proposed visual tokenizer can represent images as discrete tokens, LaVIT possesses the capability to synthesize images by auto-regressively generating visual tokens like text. We first quantitatively evaluate the model's zero-shot text-conditional image synthesis performance on the validation set of the MS-COCO benchmark (Lin et al., 2014). The detailed image generation procedure is presented in Appendix A. Following the standard setting like previous text-to-image synthesis works, we randomly sample 30k text prompts and calculate the zero-shot FID metric between real images and generated ones. The detailed comparative results are shown in Table 2. It can be seen that LaVIT outperforms all the other multi-modal language models. Compared with the concurrent work Emu, it makes a 4.3 FID improvement with a smaller LLM model, demonstrating excellent vision-language alignment capability. In addition, LaVIT achieves comparable performance with state-of-the-art text2image specialists Parti (Yu et al., 2022), while only using much fewer training data (e.g., 0.2B v.s. 2B training image-text pairs compared to Parti).

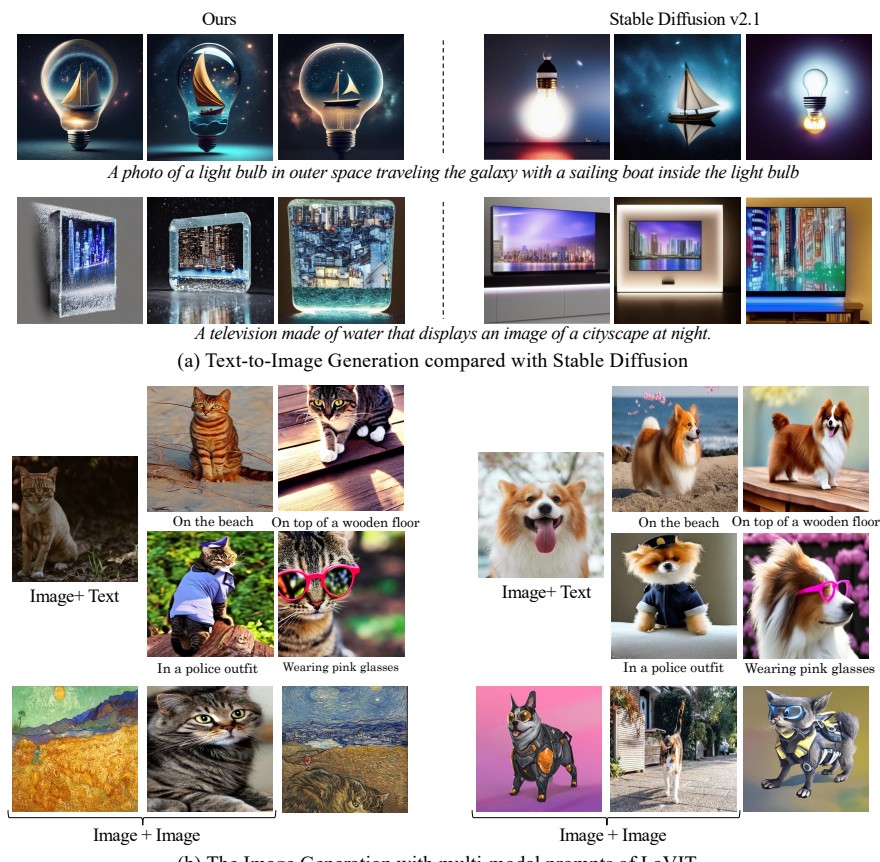

(a) Text-to-Image Generation compared with Stable Diffusion

(b) The Image Generation with multi-modal prompts of LaVIT

Figure 4: The qualitative examples of multi-modal image synthesis.

**Generation via Multi-modal Prompt** LaVIT can seamlessly accept several modality combinations (*e.g.*, text, image+text, image+image) as prompts to generate corresponding images without any fine-tuning. Figure 4 showcases some examples of the multi-modal image generation results. Our LaVIT can produce high-quality images that precisely reflect the style and semantics of the given multi-modal prompts, which demonstrates the strong multi-modal modeling potential of LaVIT. More interestingly, it can modify the original input image by the input multi-modal prompt (*e.g.*, in the last example two prompt images with a dog or cat generate a dog's portrait with the cat's whisker). This capability cannot be attained by conventional image generation models like Stable Diffusion in the absence of additional fine-tuned downstream data (Ruiz et al., 2023).

## 4.3 Ablation Study

In this study, we investigate the impact of various component designs in LaVIT on downstream performance. All the ablations were conducted on part of pre-training data by using the clip ViT-L/14 (Jia et al., 2021) as the visual encoder due to the costly training resources.

**Token Classification or Feature Regression?** When jointly training vision and language via generative training in text-oriented LLM, it is crucial to select the appropriate optimization objectives for the 2D raster-ordered visual input. When quantizing the continuous visual tokens into the discrete form, it is convenient to use the cross-entropy loss for supervising the next visual token prediction akin to textual tokens. We conjecture that such a uniform objective for both vision and language contributes to aligning them together in the LLM. To validate the superiority of the proposed visual quantization, we change the optimization objective of visual tokens to regressing the next visual embeddings by employing a regression head like Emu (Sun et al., 2023). Table 3a summarizes the results of different training objectives. As observed, adopting the regression loss for the next visual token prediction will severely degrade the model performance.

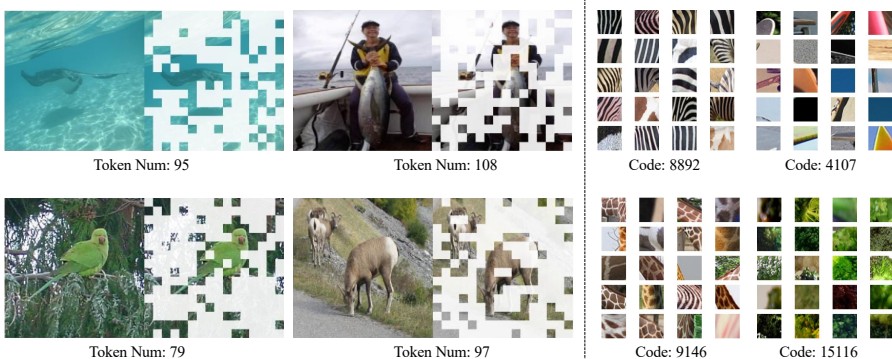

Figure 5: The visualization for the dynamic visual tokenizer (left) and learned codebook (right). Our tokenizer can dynamically select the most informative patches based on the image content and the learned codebook can produce visual codes with high-level semantics.

**Dynamic or Fixed Token Length**. Given the extracted visual features, a straightforward way is to tokenize all the patch embeddings into the visual tokens, which results in a fixed token length (i.e., 256). We compare the impact of fixed and dynamic tokenization strategies in terms of training time, computation overhead, and zero-shot performance on multi-modal understanding. As shown in Table 3b, the dynamic visual tokenizer achieves superior performance while only requiring 94 tokens on average for the input images, about 36% of the fixed one. Given that the attention computation in LLM exhibits a quadratic relationship with respect to the token length, this sparsification can accelerate the training time by 40% and reduce the computational cost in inference.

| Setting | Flickr | VQAv2 | OKVQA |
|---|---|---|---|
| Regression | 60.4 | 53.6 | 41.9 |
| Classification | **73.2** | **57.1** | **47.0** |

| Setting | Num | Time | Flickr | VQAv2 | OKVQA |
|---|---|---|---|---|---|
| Fixed | 256 | 30h | 71.1 | 56.5 | 46.4 |
| Dynamic | 94 | 18h | **74.0** | **57.7** | **47.6** |

(a) Ablations of different training objectives.    (b) Ablations for the effect of different tokenization strategies.

Table 3: The ablations of different optimization objectives for visual tokens and tokenization strategies. The num and time in Table 3b indicate the mean visual token number and pre-training time.

## 4.4 QUALITATIVE ANALYSIS

We visualize some examples processed by the proposed dynamic tokenizer. As shown in Figure 5, the token selector is capable of dynamically selecting the most informative image patches that are competent enough to represent the semantics of the whole image. Visual patches that contain repetitive or trivial background semantics are filtered during this procedure, thereby reducing redundant information and improving computing efficiency. We also visualize the image patches that belong to the same visual code in Figure 5. As observed, the learned discrete codes can convey explicit visual semantics and group the image patches with similar patterns together. For instance, code 4107 represents a part of a skateboard, and code 9146 indicates the texture of a giraffe, which strongly demonstrates the interpretability of the learned codebook.

## 5 CONCLUSION

This paper presents the LaVIT, a new general-purpose foundation model that is capable of simultaneously performing multi-modal understanding and generation. Beyond the previous adapter-based methods, it inherits the successful auto-regressive generative learning paradigm of LLMs by representing both vision and language in a unified discrete token representation via a dynamic visual tokenizer. Through optimization under the unified generative objective, LaVIT can treat images as a foreign language, comprehending and generating them like text. Extensive experimental results further demonstrate the LaVIT's superior capability to serve as a multi-modal generalist.

**Acknowledgement**: This research work is supported by National Key R&D Program of China (2022ZD0160305).

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

## A    THE DETAILS OF IMAGE SYNTHESIS

Given a multi-modal prompt (image, text, or their combinations), LaVIT first tokenizes it as a sequence of discrete tokens. By appending the special [IMG] (image start token) to the end of the prompt and feeding into the LLM, it can auto-regressively generate a sequence of visual tokens until reaches the special [/IMG] (image end). These yielded discrete visual tokens are further reconstructed into a feature map by the decoder of the proposed visual tokenizer, which reflects the high-level semantics of the synthetic image. Finally, the conditional denoising U-Net will take this feature map as the condition to progressively recover image pixels from a Gaussian noise. Taking the text prompt as an example, we illustrate the entire image synthesis procedure in Figure 6.

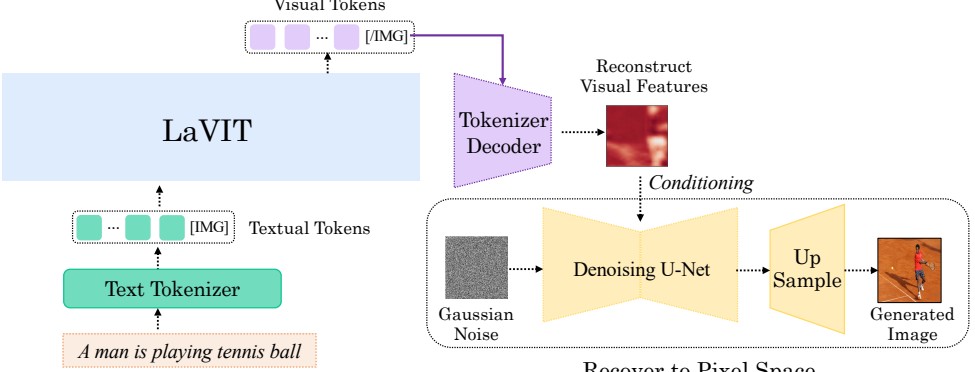

Figure 6: The detailed image synthesis process for the textual prompt by LaVIT. Generation for prompts in other modalities is similar.

During the text-to-image evaluation on MS-COCO, we use the top-k sampling to generate image tokens and set the maximum sampled tokens to $k = 300$, and softmax temperature to $1.0$. Besides, the Classifier-Free Guidance (CFG) (Ho & Salimans, 2022) strategy is also employed to enable the model to yield more prompt-aligned image tokens. Specifically, we replace the input multi-modal prompt with the [IMG] token to conduct unconditional sampling. Both conditional and prompt-conditioned visual token streams are generated in LaVIT. Hence, the real sampling logits for $t_{\text{th}}$ image token $y_t$ in the generation is a blend of both unconditional and conditional logits:

$$l(y_t) = l(y_t|[\text{IMG}]) + \alpha_{\text{cfg}} \cdot (l(y_t|\text{prompt}) - l(y_t|[\text{IMG}])) \tag{7}$$

The larger $\alpha_{\text{cfg}}$ will encourage the model to generate samples highly related to the input prompt while sacrificing some diversity. We set $\alpha_{\text{cfg}}$ to $1.5$ when evaluating the zero-shot text-to-image synthesis on MS-COCO. LaVIT generates 4 samples for each prompt and uses a CLIP-VIT/L model to select the best one according to the image-prompt matching score.

## B    ADDITIONAL EXPERIMENTAL ANALYSIS

### B.1    THE IMPACT OF VISION-LANGUAGE PRE-TRAINING ON LLM

During pre-training, the weights of LLM are unlocked to jointly optimize the visual and textual tokens. Since a new modality is included in text-oriented LLM, without setting appropriate hyper-parameters, its original language capability may be impaired. In this section, we discuss how to mitigate the forgetting of knowledge in LLM when undergoing the vision-language pre-training. We chose the massive multitask language understanding benchmark (MMLU) (Hendrycks et al., 2020) to quantitatively measure the model's common knowledge and reasoning ability. MMLU is a knowledge-intensive dataset covering various domains, including humanities, STEM, and social sciences. It is a commonly used benchmark in evaluating the Large Language Model. Following LLaMA (Touvron et al., 2023), we evaluate the model in the 5-shot setting. The detailed experimental results are reported in Table 4.

Through exploration, we conclude some useful strategies for conducting unified vision-language pre-training in LLM: 1) The Learning rate of the LLM part is the most important hyper-parameter.

| Setting | $LR_V$ | $LR_L$ | Text | Multi-modal | Humanities | STEM | Social Sciences | Other | Average |
|---------|--------|--------|------|-------------|------------|------|-----------------|-------|---------|
| LLaMA-7B | - | 3e-4 | 100% | 0% | 34.0 | 30.5 | 38.3 | 38.1 | 35.1 |
| LaVIT-7B | 1.5e-4 | 1.5e-4 | 66% | 33% | 26.7 | 26.1 | 25.7 | 29.8 | $27.1(-8.0)$ |
| LaVIT-7B | 1.5e-4 | 5e-5 | 50% | 50% | 31.4 | 27.2 | 31.8 | 34.5 | $30.9(-4.2)$ |
| LaVIT-7B | 1.5e-4 | 5e-5 | 66% | 33% | 33.5 | 30.3 | 35.2 | 36.7 | $33.6(-1.5)$ |

Table 4: The influence of learning rate and data proportion on the MMLU benchmark. Here, $LR_V$ and $LR_L$ indicate the learning rate for the visual and LLM parts in LaVIT. LaVIT learns excellent multi-modal modeling capability with only a slight drop of its original common knowledge.

| Dataset | Sampling prop |
|---------|---------------|
| C4 | 80% |
| Github | 4% |
| Wikipedia | 4% |
| Books | 4% |
| ArXiv | 4% |
| StackExchange | 4% |

Table 5: The used English text corpus proportion of different subsets in Redpajama.

Although using a large learning rate ($e^{-4}$ level) for tuning the LLM part will speed up its convergence on solving the vision-language task, it also speeds up the forgetting of its original learned common knowledge (8.0 performance drop compared to original LLaMA). To balance these, we suggest of learning rate of $e^{-5}$ level of LLM and $e^{-4}$ level of visual. 2) Text corpus data is also important. Lack of text data will also degrade the LLM's ability. Therefore, we suggest a proportion of 2:1 (L:VL) to maintain its original learned knowledge. In our experiments, the used English text corpus is from the Redpajama (Computer, 2023) dataset. The detailed proportion for each subset of Redpajama is presented in Table 5. Finally, we set the learning rate of LLM to $5e-5$ and $1.5e^{-4}$ for other parts in LaVIT, which only slightly affected LLM's language ability (-1.5 on MMLU), but achieved the best results on both multimodal understanding and generation tasks.

## B.2 MORE ABLATION STUDY

Like ablations in the main text, the following experiments are conducted on part of pre-training data by using clip ViT-L/14 (Jia et al., 2021) as the visual encoder.

**Bi-directional or Causal Attention?** The token merger in the proposed dynamic visual tokenizer includes the self-attention layer to model the interaction among the visual tokens. To convert 2D raster-ordered visual features from the ViT encoder into a sequence with causal dependency like text, causal self-attention is utilized to enforce the restriction that each visual token can solely attend to its preceding ones. To validate the effectiveness, we replace causal attention with bi-directional attention in the token merger. As shown in Table 6, causal attention outperforms bi-directional self-attention across all multi-modal understanding tasks. We conjecture the inferior performance can be attributed to partial information leakage, as the preceding token has visibility of the subsequent one, thereby impairing the optimization of next-token prediction in LLM.

| Setting | Flickr | VQAv2 | OKVQA |
|---------|--------|-------|-------|
| Bi-directional Attention | 69.9 | 55.6 | 44.1 |
| Causal Attention | **73.2** | **57.3** | **46.4** |

Table 6: The effect of different attention manners in the token merger on downstream tasks.

**Type of Visual Input Embeddings**. For multi-modal understanding where the image is treated as a condition on the left (i.e., [image, text]), the visual representations input to LLM can be quantized vectors from the codebook or the continuous features yielded by token merger. We investigate the impact of these two different visual input forms in Table 7a. As observed, the continuous visual features achieve better multi-modal understanding performance. This phenomenon is reasonable as quantized embeddings represent the centroid of continuous visual features, potentially leading to the loss of detailed visual content. Therefore, for input sequences with order [image, text], we utilize the

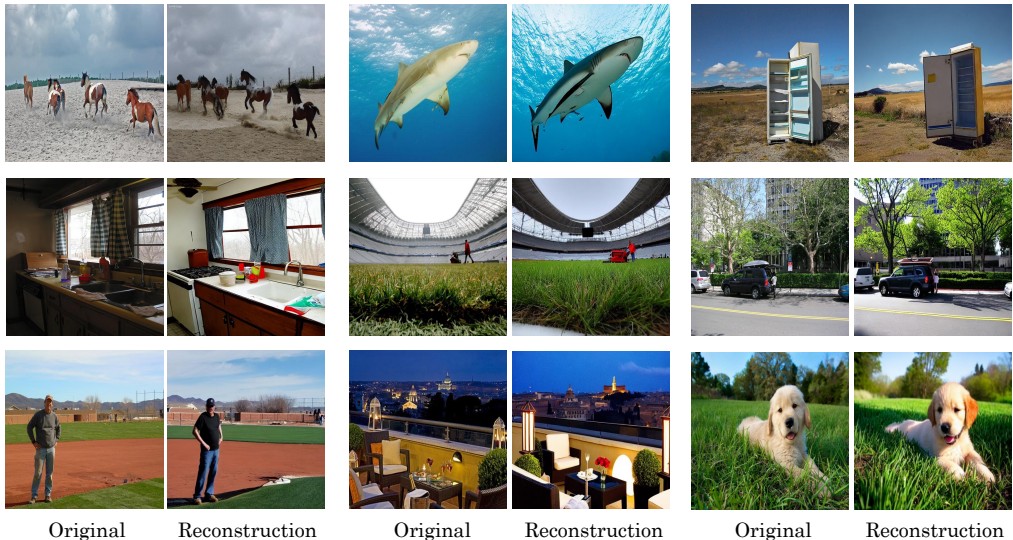

Original    Reconstruction    Original    Reconstruction    Original    Reconstruction

Figure 7: The image reconstruction results from discrete visual tokens by the denoising U-Net.

outputs from the token merger as input to LLM, whilst regarding the corresponding discrete visual codes as the supervision signal for predicting the next visual tokens.

| Setting | Flickr | VQAv2 | OKVQA |
|---|---|---|---|
| Quantized | 70.3 | 53.0 | 44.8 |
| Continuous | **72.7** | **56.4** | **46.2** |

(a) The effect of visual input embeddings forms.

| Setting | Flickr | VQAv2 | OKVQA |
|---|---|---|---|
| Frozen LLM | 63.0 | 52.7 | 44.3 |
| Unlock LLM | **72.1** | **57.1** | **47.9** |

(b) The effect of unlocking LLM in training.

Table 7: The ablation studies of visual input embeddings format and the unlock of LLM during pre-training. The performance is evaluated on the zero-shot setting.

| Setting | Flickr | VQAv2 | OKVQA |
|---|---|---|---|
| w/o Token Merger | 70.8 | 51.9 | 42.1 |
| w/ Token Merger | **72.7** | **56.4** | **46.2** |

Table 8: The effect of the proposed token merger on downstream multi-modal understanding tasks.

**The role of Token Merger**. The token merger aims to compress the semantics of unselected visual patches onto the retained ones according to their feature similarity. Compared to directly discarding these unselected patches, this merge mechanism not only reduces the redundancy among patches but also maximally preserves the visual details. The ablation study to explore the role of our token merger is shown in Table 8. From the presented results, it can be observed a distinct performance drop in multi-modal understanding tasks when eliminating the token merger module.

**The Effect of Unlock LLM**. In order to ascertain the efficacy of utilizing LLM for learning the interaction between vision and language, we devise an ablation experiment wherein the entirety of LLM remains frozen while solely optimizing the parameters of the token merger during training. The detailed comparison results are shown in Table 7b. One can observe that freezing the language model will restrict its powerful multi-modal modeling potential and thus result in a noticeable performance degradation. Hence, unlocking the LLM under a unified training objective will contribute to adequately bridging the semantics between vision and language.

## C  MORE VISUALIZATIONS

**The quality of pixel decoding**  In Figure 7, we visualize some pixel decoding examples by the trained denoising U-Net. It can be seen that, given the tokenized discrete visual tokens, the original input images can be successfully recovered. The reconstructed examples exhibit a high degree of

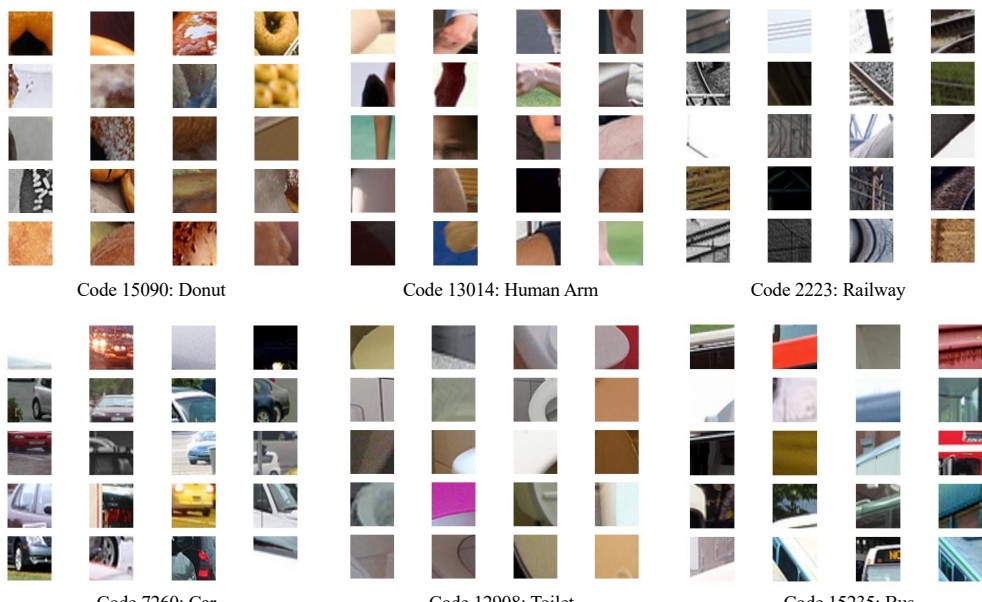

Code 15090: Donut          Code 13014: Human Arm          Code 2223: Railway

Code 7260: Car          Code 12908: Toilet          Code 15235: Bus

Figure 8: The visualization for the leaned discrete visual codes.

| Configuration | VL Pre-training | Tokenizer training |
|---|---|---|
| Visual Encoder | EVA-CLIP ViT-G/14 | EVA-CLIP ViT-G/14 |
| LLM init | LLaMA-1-7B | - |
| Optimizer | AdamW | AdamW |
| Optimizer Hyperparameters | $\beta_1 = 0.9, \beta_2 = 0.95, \epsilon = 1e^{-6}$ | $\beta_1 = 0.9, \beta_2 = 0.99, \epsilon = 1e^{-6}$ |
| Global batch size | 2048 | 2048 |
| Peak learning rate of LLM | 5e-5 | - |
| Peak learning rate of Visual Part | 1.5e-4 | 2e-4 |
| Learning rate schedule | Cosine | Cosine |
| Training Steps | 20K | 50K |
| Warm-up steps | 2k | 4K |
| Weight decay | 0.1 | 0.01 |
| Gradient clipping | 1.0 | 1.0 |
| Input image resolution | 224 * 224 | 224 * 224 |
| Input sequence to LLM | 2048 | - |
| Numerical precision | bfloat16 | bfloat16 |
| GPU Usage | 256 NVIDIA A100 | 64 NVIDIA A100 |
| Training Time | 30h | 12h |

Table 9: The detailed training hyperparameters of LaVIT

semantic and general structure to the original images. This consistency validates that our dynamic tokenizer can efficiently encode visual information using generated discrete tokens.

**The Learned Codebook**   We also provide more visualization examples for the learned codebook in Figure 8, where the image patches belonging to the same discrete visual code are arranged together. One can observe that the learned visual code can encode explicit high-level visual semantics. For example, code 13014 represents the human arm, code 7260 shows the part of the car, code 15235 represents the bus, and code 15090 indicates the donuts. The high distinctiveness of the learned codebook facilitates the unified vision-language pre-training in LLM.

**Multi-Modal Image Synthesis**   We illustrate more image synthesis results in Figure 9, 10, 11 and 12. As presented, LaVIT produces high-quality, content-rich, and appearance-diverse image samples that precisely reflect the prompt's semantics. Importantly, LaVIT can produce images well aligned with long and more complex text descriptions, while the samples generated by stable diffusion do not reflect the subtle details in some complicated prompts. Besides, LaVIT also supports the multi-modal in-context image generation. As illustrated in Figure 9, the synthetic samples can

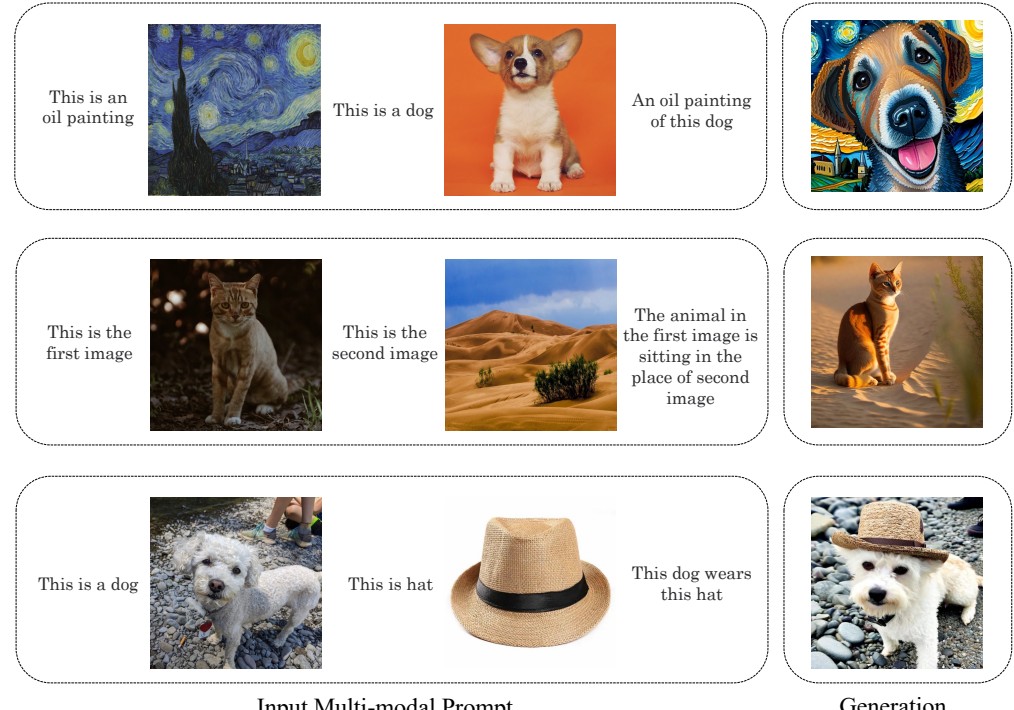

Figure 9: The examples of multi-modal in-context image synthesis.

fully reflect the semantics of input multi-modal context, which contributes to the diversified editing of the input image through the multi-modal control.

**Multi-modal Understanding**    We also visualize some multi-modal understanding examples in Figure 13. As observed, LaVIT is capable of understanding the visual content, generating concise textual captions to depict the image, and answering diverse questions about detailed visual clues.

## D    PRE-TRAINING DETAILS

The detailed training hyper-parameter settings are reported in Table 9.

## E    LIMITATIONS

Since LaVIT is based on the Large Language Model, it inherits LLM's original language hallucination limitations, e.g., generating some nonexistent knowledge and making some factual errors. Besides, LaVIT is trained only on web-scale image-text pairs, wherein the text descriptions are noisy and short, thereby rendering LaVIT insufficient for effectively modeling extensive text-image correspondences. We believe this limitation can be alleviated when leveraging more high-quality and long image-text interleaved documents.

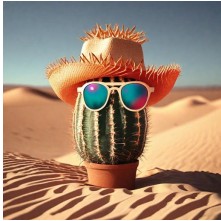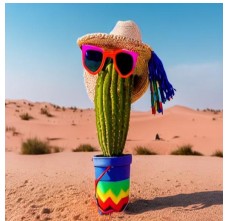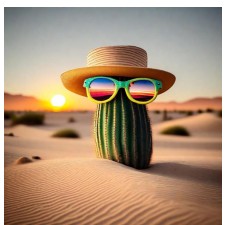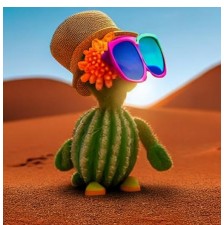

*A small cactus wearing a straw hat and neon sunglasses in the Sahara desert.*

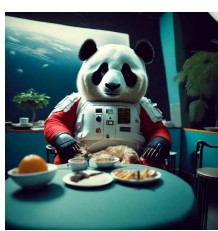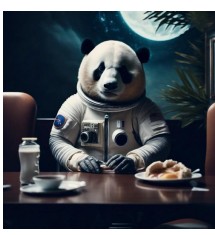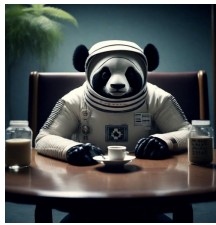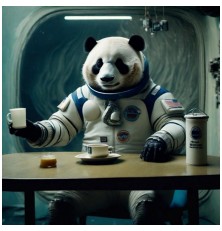

*A high contrast photo of panda dressed as an astronaut sits at a table in a photorealistic style*

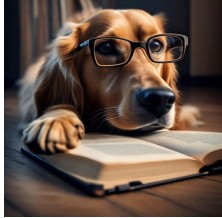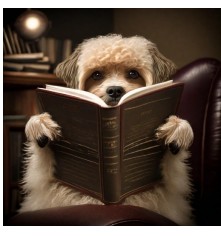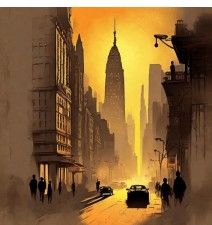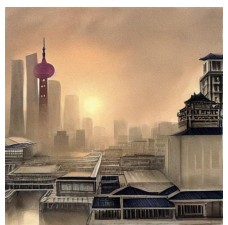

*A lovely dog is reading a thick book*      *Downtown Shanghai at sunrise. Detailed ink wash.*

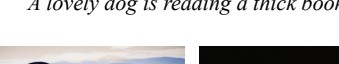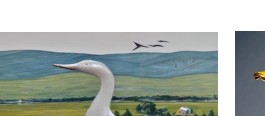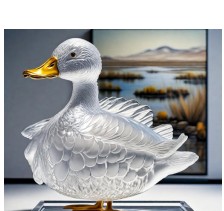

*A transparent sculpture of a duck made out of glass. The sculpture is in front of a painting of a landscape*

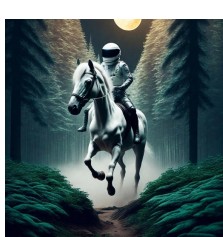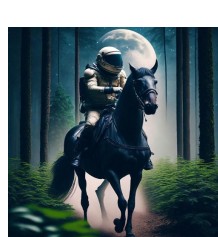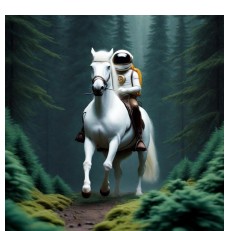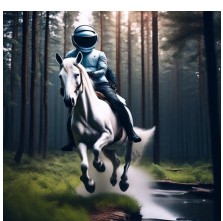

*A high contrast photo of an astronaut riding a horse in the forest.*

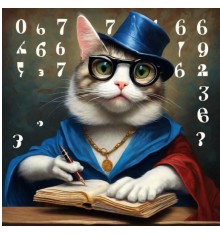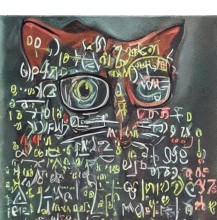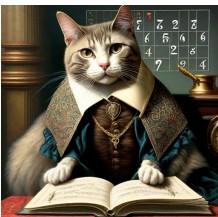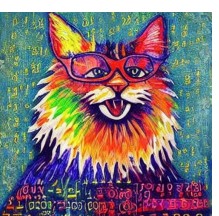

*a super math wizard cat, richly textured oil painting*

Figure 10: The examples of text-to-image synthesis.

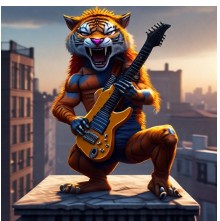 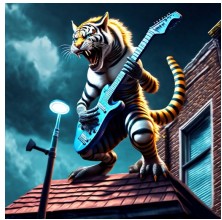 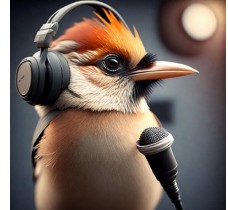 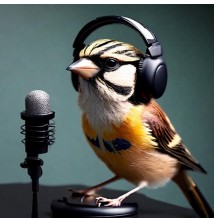

*A heavy metal tiger standing on a rooftop while singing on an electric guitar under a spotlight.*     *a bird wearing headphones and speaking into a high-end microphone in a recording studio.*

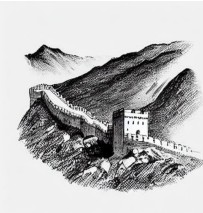 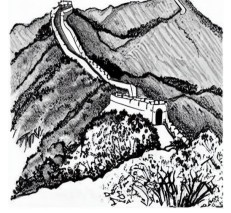 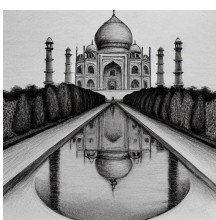 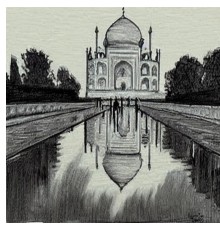

*A section of the Great Wall in the mountains. detailed charcoal sketch*     *Taj Mahal with its reflection. detailed charcoal sketch.*

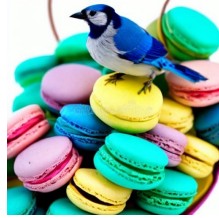 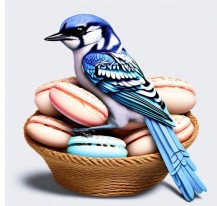 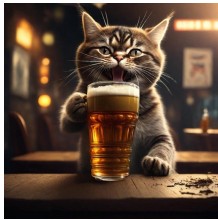 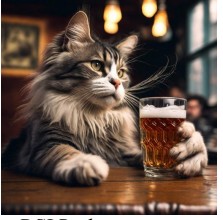

*A blue jay standing on a large basket of rainbow macarons.*     *a cat drinking a pint of beer, DSLR photo, art station, octane rendering*

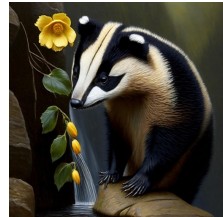 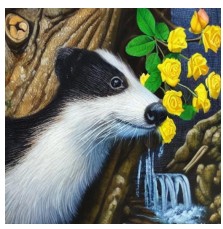 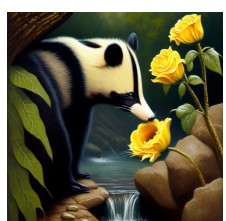 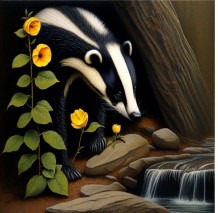

*A richly textured oil painting of a young badger delicately sniffing a yellow rose next to a tree trunk. A small waterfall can be seen in the background.*

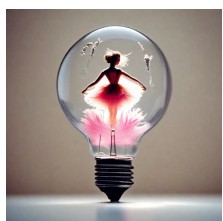 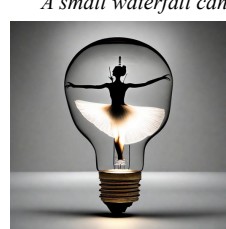 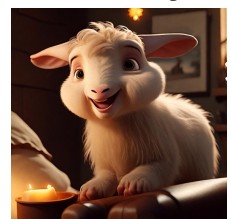 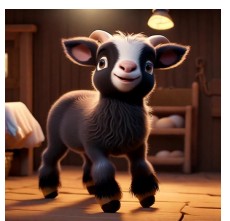

*a light bulb with a ballerina in it*     *Cute adorable little goat, unreal engine, cozy interior lighting, art station, detailed digital painting*

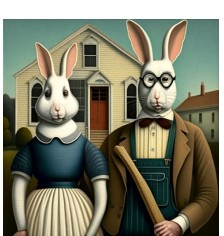 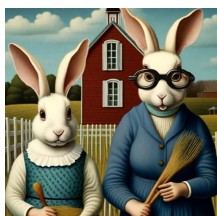 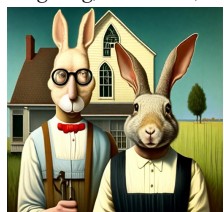 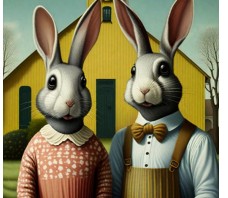

*A painting of two rabbits in the style of American Gothic, wearing the same clothes as in the original*

Figure 11: The examples of text-to-image synthesis.

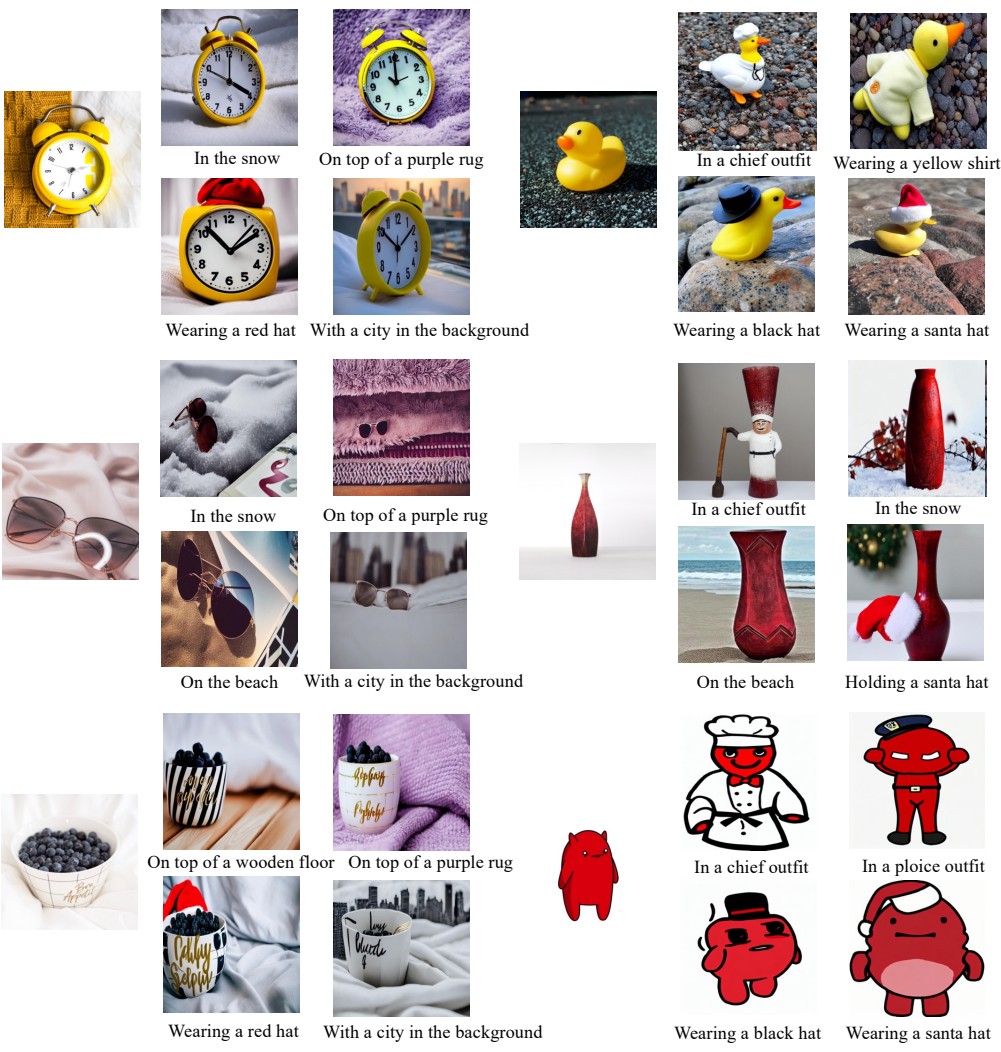

Figure 12: The examples of image synthesis with multi-modal prompt.

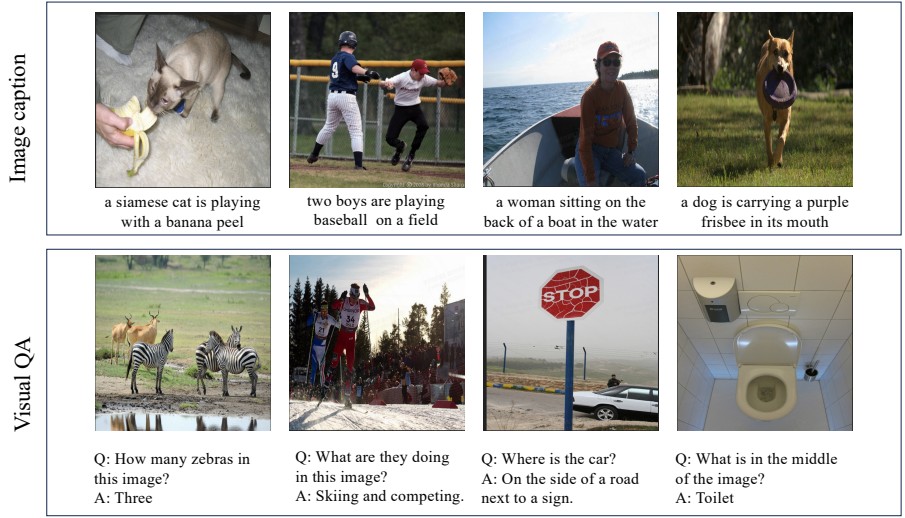

Figure 13: The qualitative evaluation examples on model's multi-modal understanding performance.

