# OpenReview forum: "Unified Language-Vision Pretraining in LLM with Dynamic Discrete Visual Tokenization"
_ICLR.cc/2024/Conference — ICLR 2024 poster_

### Official Review · Reviewer_54hM · 2023-10-17

**Soundness:** 3 good
**Presentation:** 2 fair
**Contribution:** 3 good
**Rating:** 6
**Confidence:** 4

**Summary:**

This paper presents a vision-language framework with unified objectives for both language and image.
The crux of the proposed method is the design of a dynamic discrete visual tokenization module.
In particular, the proposed LaVIT leverages a dynamic token sparsification approach to embed the given image with fewer vision tokens.
After that, the learned tokenizer is employed to generate discrete vision tokens.
In this way, the proposed framework can achieve a unified pre-training objective.

**Strengths:**

- The proposed method achieves very impressive performance on several vision-language tasks such as VQA and image captioning.
- The strategy to pre-train the vision-language models with a unified objective seems interesting.
- The method can generate high-quality images based on the given textual prompt.

**Weaknesses:**

- The structure of the writing, especially the method section, is very confusing to me.
In fact, the key to the proposed method is the learning of the vision token codebook.
This codebook and token selector are built on the first stage of learning.
However, the authors' description of this part makes it look like an end-to-end training framework, which confuses me a lot.
I would suggest the authors carefully revise the structure of this section.

- The authors claim their method can achieve in-context learning capability.
However, I cannot find any experiments pertaining to this merit.

- The authors did not provide a good explanation of why inequitable treatment for language and vision harms multi-modal learning.

- More details about the language-vision pre-training part are required.

**Questions:**

- Is it possible that the impressive performance of the proposed method is due to more training data, especially for the vision tokenization codebook?

- Why do the authors still use the unselected tokens as keys and values?
One intuitive approach for these tokens is simply discarding them.
Moreover, these tokens should contribute minorly to the final performance and are redundant as authors claimed.

- For the causal attention, which tokens are seen as previous ones?
As there is actually no sequence for an image, using causal attention for image encoding is thus not convincing.

- What is the function of the second half of Eqn. 4?

---

> ### Author Response · Authors · 2023-11-17
> **Response to Reviewer 54hM - Part1**
>
> We sincerely appreciate the time and effort you dedicated to reviewing our paper and providing valuable feedback. Below are our responses to the raised questions.
>
>
> 1. *The structure of the writing, especially the method section, is very confusing to me.  In fact, the key to the proposed method is the learning of the vision token codebook. ...  I would suggest the authors carefully revise the structure of this section.*
>
> * Sorry for the confusion caused by our description of the methodology section. We will make further revisions to the presentation of this section based on your constructive suggestions. We would appreciate it if you have any other detailed revision suggestions.
>
> 2. *The authors claim their method can achieve in-context learning capability. However, I cannot find any experiments pertaining to this merit.*
>
> * Thanks for pointing out this. The "in-context capability" mentioned in this work means that the model can take multi-modal prompt input as context and successfully generate context-related contents. We have evaluated LaVIT’s few-shot in-context learning ability on the downstream multi-modal understanding benchmarks. Specifically, we sample several examples from the training data to make task-specific prompts in the form of interleaved [image, text]. For a fair comparison, the few-shot samples are selected by employing the Retrieval In-Context Example Selection (RICES) strategy [3] following the existing methods [1,2]. As observed, LaVIT can learn to complete specific tasks given only a few examples in the form of demonstration. Furthermore, the performance of LaVIT will improve as the number of support samples increases. This phenomenon is consistent across the three benchmarks, demonstrating its remarkable multi-modal in-context learning potential.
>
> | Method |  | VQAv2 |  | | Vizwiz |  |
> | :-----:| :----: | :----: | :----: | :----:|:----: |:----: |
> |  | k=2 | k=4 | k=8 | k=2 | k=4 | k=8 |
> | KosMos-1 | 51.4 | 51.8 |  51.4 | 31.4 | 35.3 | 39.0 |
> | Flamingo-9B | - | 56.3 | 58.0 | - |  34.9 | 39.4 |
> | Emu | 56.4 | 58.4 | 59.0 | 37.8 | 41.3 | **43.9** |
> | Ours | **66.5** | **67.2** | **67.8** | **38.5** | **41.7** | 42.1 |
>
>
> * We also provide more visualization examples of the multi-modal in-context image synthesis results in Figure 9 of the appendix. It can be seen that LaVIT is capable of understanding the semantics of multi-modal prompts and generating the context-related images without any other fine-tuning.
>
> 3. *The authors did not provide a good explanation of why inequitable treatment for language and vision harms multi-modal learning.*
>
> * The training objective of most existing multi-modal large language models is centered on predicting the textual descriptions dependent on visual content, where the visual input is merely regarded as the prompts. Since there is no supervision for visual modality input, the model only learns to generate text based on images and can not possess the capacity to generate images like language. Therefore, we claim that "the inequitable treatment of different modal inputs severely constrains the model’s potential, limiting it to only performing comprehension tasks like generating text based on images" (instead of "harms multi-modal learning"). We hope this clarification will address your concerns.
>
>
> 4. *More details about the language-vision pre-training part are required.*
>
> * The detailed training hyperparameter settings are reported in Table 9 of our appendix. Please let us know if there are any training details that we have not expressed clearly.
>
> 5. *Is it possible that the impressive performance of the proposed method is due to more training data, especially for the vision tokenization codebook?*
>
> * For a fair comparison, we employ the same multi-modal pre-training data as BLIP-2 to exclude the performance gains from additional training data. All the used training data (including about 93M samples for understanding and 100M samples for image generation) are from the publicly available datasets introduced in Section 3.3. This data scale is smaller than that of most other methods. For example, the concurrent work EMU use 2.6B data, KOSMOS adopts 3.1B image-caption pairs, and CM3Leon-7B leverage 340M closed-source data. In comparison, our LaVIT has achieved better performance while utilizing relatively less multi-modal data. Besides, the visual codebook is trained using pure images from the same multi-modal data mentioned above without the corresponding captions. This stage does not require additional data.

---

> ### Author Response · Authors · 2023-11-17
> **Response to Reviewer 54hM - Part2**
>
> 6. *Why do the authors still use the unselected tokens as keys and values? One intuitive approach for these tokens is simply discarding them. Moreover, these tokens should contribute minorly to the final performance and are redundant as authors claimed.*
>
> * Admittedly, merely using the selected tokens can approximately encode the global semantics of the image content. However, in our experiments, we have found that directly discarding the unselected patches may result in some information loss, such as the semantics of some background objects, which may impact the performance of multimodal understanding. For instance, in visual question answering tasks, if the question refers to these background objects, the model may be unable to provide the correct answer. Therefore, we design the token merger to compress the semantics of unselected patches onto the retained ones instead of directly discarding them. This process is fulfilled according to the semantic similarity between these patches, so the unselected tokens serve as keys and values. We also conducted the ablation study to explore the role of token merger. As shown by the below Table, we can observe a distinct performance drop in multi-modal understanding tasks without the token merger. In summary, the combination of token selector and merger contributes to reducing the redundancy among tokens while maximally preserving the image details.
>
> | Setting  | Flickr | VQAv2 | OKVQA | GQA |
> | :-----:| :----:|:----: |:----: |:----: |
> | w/o Token Merger | 70.8 |  51.9  |  42.1 |  39.7  |
> | w/ Token Merger | **72.7**   |  **56.4**   | **46.2** | **42.3** |
>
>
> 7. *For the causal attention, which tokens are seen as previous ones? As there is actually no sequence for an image, using causal attention for image encoding is thus not convincing.*
>
> * In the causal attention, the remaining merged visual tokens are flattened into the 1D sequence in the raster order, where the previous tokens of each one are originally on its top left side. Though the original input image is 2D shaped, the usage of causal attention will contribute to converting 2D visual features into a 1D sequence with causal dependency in the output space of token merger. To validate the effectiveness, we replace causal attention with bi-directional attention in the token merger. The results are shown in Table 6 of the appendix. Compared to bi-directional attention, adopting causal attention achieves superior performance across all multi-modal understanding tasks. This phenomenon is reasonable as the next-token prediction objective of LLM requires the causal dependency between input tokens, using causal attention will ensure the consistency with the LLM.
>
> | Setting  | Flickr | VQAv2 | OKVQA |
> | :-----:| :----:|:----: |:----: |
> | Bi-directional Attention | 69.9 |  55.6  |  44.1 |
> | Causal Attention | **73.2** | **57.3** | **46.4** |
>
>
> 8. *What is the function of the second half of Eqn. 4?*
>
> * The second half of Eq-4 serves as a regularization term to impose constraints to minimize the number of retained tokens. We expect to preserve the whole semantic information of an image with as few tokens as possible.
>
> Thank you again for your insightful reviews and valuable suggestions. Please let us know if there are further questions.
>
> ---
>
> [1] Quan Sun, et al. Generative pretraining in multimodality, arXiv preprint 2023
>
> [2] Jean-Baptiste, et al. Flamingo: a visual language model for few-shot learning, NeurIPS 2022
>
> [3] Zhengyuan, et al. An empirical study of gpt-3 for few-shot knowledge-based vqa, AAAI 2022

---

### Official Review · Reviewer_L5Te · 2023-10-22

**Soundness:** 2 fair
**Presentation:** 3 good
**Contribution:** 2 fair
**Rating:** 5
**Confidence:** 5

**Summary:**

This paper proposes the idea of treating images and text equally, transforming images into a series of discrete tokens to input into LLM, and training LLM from scratch to complete understanding and generation. The experimental part of the paper yielded so-called advanced results.

**Strengths:**

1. The paper proposes to convert the image into a series of discrete tokens and adjust the token length according to the image content.
2. The visualization of discrete tokens is displayed in Experiments to help readers understand the semantic content carried by tokens.
3. Better results than the baseline method are achieved in the experiment.

**Weaknesses:**

1. The novelty of this paper is limited and the ideas of the paper have already been explored in previous work. For example, end-to-end training LLM has been explored in Emu[MR1], and the transformation of visual information into discrete tokens have been explored in [MR2, MR3, MR4, MR5].

2. The generation process relies on a stable diffusion model, which makes the verification of the visual generation capabilities of the proposed model limited. Do the image generation results rely on strong stable diffusion, or are they more attributable to the discrete labels generated by the model? It is unclear to what extent the good results achieved are due to stable diffusion.

3. Visual token merger part is not well validated. How much additional complexity does this stage bring to the model? Are different codebooks needed for data in different domains (animation, painting, natural landscape, etc.)? Why is a complex merger mechanism needed? Can a simple attention map from a pre-trained model indicate the importance of individual image patches?

3+. BTW, the processing of the decoder (using the stable diffusion model) is the same as Emu's [MR1] strategy. A reference to the corresponding location and some justifications are required.

4. In the experimental results in Table 3, using a larger number of tokens (fixed) achieves worse results, and more explanations need to be added.




[MR1] Generative Pretraining in Multimodality. 2023.
[MR2] Image as a Foreign Language: BEIT Pretraining for Vision and Vision-Language Tasks. 2023.
[MR3] Visual Transformers: Token-based Image Representation and Processing for Computer Vision. 2020.
[MR4] Learning to Mask and Permute Visual Tokens for Vision Transformer Pre-Training. 2023.
[MR5] Beit v2: Masked image modeling with vector-quantized visual tokenizers. 20022.

**Questions:**

Refer to weakness.

---

> ### Author Response · Authors · 2023-11-17
> **Response to Reviewer L5Te - Part1**
>
> We sincerely appreciate the time and effort you dedicated to reviewing our paper and providing valuable feedback. Below we clarify the raised concerns one by one.
>
> 1. *The novelty of this paper is limited and the ideas of the paper have already been explored in previous work. ... has been explored in Emu[MR1], and the transformation of visual information into discrete tokens have been explored in [MR2, MR3, MR4, MR5]*
>
> We would clarify the unique technical contributions of LaVIT as follows:
> * (1) This work mainly focuses on exploring what makes a good visual token design when extending text-oriented LLM to deal with multimodal inputs. Although the vector quantization for visual features was explored in some previous works [2,3,4,5], directly applying these codebook techniques to tokenize images into discrete tokens trained with LLM are sub-optimal. As a key difference from textual tokens, visual tokens (image patches) exhibit a notable interdependence. In other words, the mutual information between different visual tokens is bigger compared to text tokens, making the semantics of a visual token can be deduced from its adjacent patches. This is also why the majority of mask image modeling methods [4,5] employ an exceptionally high mask rate for image patches. Analogously, such visual interdependence will render the next-token paradigm less effective through self-supervision. Hence, the proposed dynamic visual tokenizer introduces the token selector and merger to pick the most informative tokens and merge the trivial ones. This mechanism contributes to reducing the redundancy and interdependence among visual patches, rendering the unified next-token prediction objective more effective. We have conducted experiments about different visual tokenization techniques on the same 30M pre-training data to validate this. As observed, these methods are not specially tailored for LLM and thus achieve inferior performance compared to LaVIT.
>
> | Method | Codebook Size |  Flickr | GQA | VQAv2 | OKVQA |
> | :-----:| :----: |  :----: | :----:|:----: |:----: |
> | VQ-VAE | 8192 | 60.5 |  35.0  | 48.8 | 40.2 |
> | BEIT-2 | 8192 | 65.8  | 38.3 | 51.1 | 41.4 |
> | Ours | 8192 | **70.0** |  **40.8**  | **53.0** | **44.5** |
>
> * (2) The developed visual tokenizer can produce tokens with a dynamic sequence length varying from the complexity of the image content itself. However, the existing methods tokenize the image into a long and fixed sequence length (e.g., 512 or 1024). Given that the attention computation in LLM exhibits a quadratic relationship with respect to the token length, a long sequence will invariably reduce the computation efficiency. The tokens sparsification of LaVIT's tokenizer thus enables efficient multi-modal training and inference (as shown in Table 3 (b)).
>
> * (3) The same discrete format for two modalities' tokens contributes to the unification of both training and inference. For training, both visual and textual tokens can be optimized under a unified next-token classification objective indiscriminately. In inference, the LaVIT can take any modality combinations as input and produce both visual and textual content by generative sampling. Although the concurrent work Emu [1] also proposes to unlock the LLM during pre-training, it optimizes the visual modality by regressing the next visual features. As validated by Table 3 (a) in Section 4.3, the inconsistent optimization objectives for image and text are not conducive to unified multi-modal modeling within LLM. Therefore, our LaVIT can achieve superior performance than Emu using less training data and smaller model size.

---

> > ### Comment · Reviewer_L5Te · 2023-11-20
> > **Feedback**
> >
> > Got it. Thanks the authors for answering my question.
> >
> > I read some baseline papers again, and, I find another question. Could the authors give some information about the dataset of different methods?

---

> > > ### Author Response · Authors · 2023-11-20
> > > **Replying to Reviewer L5Te**
> > >
> > > The detailed comparison of pre-training datasets used in different methods is as follows:
> > >
> > > * Flamingo:
> > >     * **Interleaved image and text data**: M3W (43 million webpages)
> > >     * **Image-Text Pairs**: ALIGN (1.8B), LTIP (312M)
> > >     * **Video-Text Pairs**: VTP (27M)
> > >
> > > * Kosmos:
> > >     * **Image-Text Pairs**: LAION-2B [1],  LAION-400M [2], COYO-700M [3], Conceptual-Captions (14M) [4]
> > >     * **Interleaved Image-Text Data**: 71M webpages
> > > * Emu:
> > >     * **Image-Text Pairs**: LAION-2B [1], LAION-COCO (600M) [6], LAION-Aesthetics [9] (A high-aesthetics image subset of LAION-2B [1])
> > >     * **Video-text Pairs**: WebVid-10M [7]
> > >     * **Interleaved Image-Text Data**: MMC4 (400M images) [8]
> > >     * **Interleaved Video and Text**: YT-Storyboard-1B [10]
> > >
> > > * BLIP-2:
> > >     * **Image-Text Pairs**: MS-COCO (0.5M), Visual Genome (0.8M), Conceptual-Captions(14M) [4], SBU (1M) [5], 115M samples from the LAION400M [2]
> > >
> > > * LaVIT (Ours):
> > >     * **Image-Text Pairs**: Conceptual-Captions(14M), SBU (1M), 115M samples from the LAION400M
> > >     * 100M samples from LAION-Aesthetics [9] (A high-aesthetics image subset of LAION-2B [1]) for training text-to-image generation.
> > >
> > > Of these, Flamingo uses closed-source datasets, while the others use open-source datasets. From the above comparison, it can be seen that LaVIT uses a smaller scale of pre-training data compared to existing methods. All the compared baselines are tested in the same evaluation dataset.
> > >
> > > Besides, we also adopted the same training data and language model (LLaMA-7B) in our ablation study to compare with BLIP2 and Emu, following their released official implementation. The comparison results are reported as follows. We hope these responses have settled your questions.
> > >
> > >
> > > | Setting  | Nocaps |Flickr | VQAv2 | OKVQA |
> > > | :-----:| :----:|:----:|:----: |:----: |
> > > | Emu | 98.5 | 60.4 | 53.6  |  41.9 |
> > > | BLIP2 | 101.3  | 68.5 | 53.0  | 42.8 |
> > > | Ours | **106.2**  | **73.2** | **57.1** | **47.0** |
> > >
> > > ---
> > >
> > > [1]  Christoph Schuhmann, et al. Laion-5b: An open large-scale dataset for training next generation image-text models
> > >
> > > [2]  Christoph Schuhmann, et al. Laion-400m: Open dataset of clip-filtered 400 million image-text pairs
> > >
> > > [3] Minwoo Byeon, et al. Coyo-700m: Image-text pair dataset, 2022
> > >
> > > [4] Soravit Changpinyo, et al. Conceptual 12m: Pushing web-scale image-text pre-training to recognize long-tail visual concepts.
> > >
> > > [5] Ordonez, V, et al. Im2text: Describing images using 1 million captioned photographs.
> > >
> > > [6] Laion coco: 600m synthetic captions from laion2b-en. https://laion.ai/blog/laion-coco/.
> > >
> > > [7] Max Bain, et al. Frozen in time: A joint video and image encoder for end-to-end retrieval.
> > >
> > > [8] Wanrong Zhu, et al. Multimodal c4: An open, billion-scale corpus of images interleaved with text.
> > >
> > > [9] Laion-aesthetics. https://laion.ai/blog/laion-aesthetics/.
> > >
> > > [10] Rowan Zellers, et al. Merlot reserve: Neural script knowledge through vision and language and sound.

---

> ### Author Response · Authors · 2023-11-17
> **Response to Reviewer L5Te - Part2**
>
> 2. *The generation process relies on a stable diffusion model, which makes the verification of the visual generation capabilities of the proposed model limited. Do the image generation results rely on strong stable diffusion ... It is unclear to what extent the good results achieved are due to stable diffusion.*
>
> * The denoising U-Net in LaVIT serves as a pixel decoder to recover image pixels (rendering) from the corresponding discrete visual token. In other words, this U-Net is responsible for mapping from discrete space to pixel space rather than generating new image contents based on input prompts. As shown in Figure 7 of the appendix, the reconstructed images from token IDs by the U-Net exhibit a high degree of semantics and general structure to the original images. Therefore, the quality of the generated image (i.e., prompt-following, aesthetic, diversity) is predominantly determined by the discrete visual token generated by the capability of LaVIT rather than this decoder. This claim is further supported by evidence from the following two aspects:
>
>
> * (1) Despite the weight of conditional U-Net in LaVIT is initialized from the Stable Diffusion v1.5, the LaVIT achieves the superior text-to-image generation performance than the SD v1.5 (7.4 (Ours) v.s. 9.9 (SD 1.5) in FID metric as in Table 2). Hence, this improvement stems from the LaVIT's capability to better correlate the visual and textual tokens.
>
> * (2) Both GILL[6] and Emu[1] have leveraged the same pre-trained weight of Stable Diffusion v1.5 like ours as the weight initialziation for the decoder. After training on the same data (i.e., the open-source LAION-Aesthetics), they still achieved inferior performances than LaVIT (e.g., 7.4 (Ours) v.s. 12.2 (GILL) and 11.7 (Emu)) as shown in Table 2.
>
> * Therefore, all the above evidence can fully demonstrate that the good zero-shot image generation results are attributed to the strong capability of LaVIT.
>
>
> 3. *Visual token merger part is not well validated. ... Why is a complex merger mechanism needed? Can a simple attention map from a pre-trained model indicate the importance of individual image patches?*
>
> * **Why is a complex merger mechanism needed?** The token merger aims to compress the semantics of unselected visual patches onto the retained ones according to their feature similarity. Compared to directly discarding these unselected patches, this merge mechanism not only reduces the redundancy among patches but also maximally preserves the visual details. The ablation study to explore the role of token merger is shown as follows. As observed, the introduction of this module can significantly improve the performance of multimodal understanding.
>
> | Setting  | Flickr | VQAv2 | OKVQA | GQA |
> | :-----:| :----:|:----: |:----: |:----: |
> | w/o Token Merger | 70.8 |  51.9  |  42.1 |  39.7  |
> | w/ Token Merger | **72.7**   |  **56.4**   | **46.2** | **42.3** |
>
> * **attention map from a pre-trained model.** Although the attention map from the pre-trained ViT can reflect the importance of individual image patches to some extent, it is difficult to manually determine a threshold based on attention weights to select which tokens should remain as input to LLM. In contrast, the proposed token select and merge mechanism can dynamically pick the most informative visual tokens based on the complexity of the image content itself.
>
> * **Additional complexity.** The token merger consists of 12 transformer blocks. It has approximately 280 million parameters, comprising just 3% of the total model. This allocation of parameters does not significantly increase the complexity of the model.
>
> * **Different codebooks needed.** Our LaVIT does not require different codebooks for data from different domains. The learned codebooks can be adapted to diverse domains. As shown in Figures 10 and 11 of the appendix, LaVIT can generate images with different styles, such as oil painting, DSLR photos, animation, and landscape.

---

> ### Author Response · Authors · 2023-11-17
> **Response to Reviewer L5Te - Part3**
>
> 4. *BTW, the processing of the decoder (using the stable diffusion model) is the same as Emu's [MR1] strategy. A reference to the corresponding location and some justifications are required*
>
> * Thank you for pointing out the lack of discussion with the Emu decoder. There are the following main differences between these two decoders. (1) The Emu's decoder takes the continuous visual embeddings yielded by LLM as the conditions to generate images. In contrast, our pixel decoder is conditioned on the discrete visual token IDs. It contains an additional module to recover the dynamic discrete token sequence back to the corresponding visual semantic features $x_{\text{rec}}$. The pixel decoding procedure of LaVIT is based on the reconstructed $x_{\text{rec}}$. (2) The training strategies for two decoders are different. the training of Emu's decoder requires the image-text pairs and LLM, during which text is first fed into LLM to generate the visual embeddings. However, our decoder only takes pure image data as input and does not need the combination with LLM during training (only at first stage of training), which reduces the computational overhead and the requirement of the rare multi-modal training data. We have added these discussions in the appendix following your valuable suggestions.
>
>
> 5. *In the experimental results in Table 3, using a larger number of tokens (fixed) achieves worse results, and more explanations need to be added.*
>
> * The fixed tokenizer variant transforms all the image patch features into the visual tokens as input to LLM during pre-training.  As mentioned above, the redundancy and high interdependence between the image patches may affect the effectiveness of next-token prediction through self-supervision. In comparison, the proposed dynamic tokenizer can reduce the trivial visual redundancy and thus achieve superior zero-shot performance.
>
> Thank you again for the detailed reviews of our paper. We hope these responses have settled your concerns. Please let us know if there are further questions or concerns.
>
> ---
>
> [1] Quan Sun, et al. Generative pretraining in multimodality, arXiv preprint 2023
>
> [2] Esser, Patrick, et al. Taming transformers for high-resolution image synthesis, CVPR 2021
>
> [3] Aditya Ramesh, et al. Zero-shot text-to-image generation, ICML 2021
>
> [4] Wenhui Wang, et at. Image as a Foreign Language: BEIT Pretraining for Vision and Vision-Language Tasks. CVPR 2023
>
> [5] Peng, Zhiliang, et al. Beit v2: Masked image modeling with vector-quantized visual tokenizers, arXiv preprint 2022
>
> [6] Jing, et al. Generating images with multimodal language models, arXiv preprint 2023

---

### Official Review · Reviewer_K5nK · 2023-10-31

**Soundness:** 3 good
**Presentation:** 3 good
**Contribution:** 3 good
**Rating:** 8
**Confidence:** 3

**Summary:**

This paper proposes an image tokenization method for visual-language model (VLM) pretraining, called dynamic discrete visual tokenization. The proposed method has two key characteristics:
* Discrete: Continuous feature vectors from each image patch are mapped to a learnable codebook, resulting in discrete visual tokens compatible with text tokens. The discrete visual tokens make it possible to pre-train VLMs using the same autoregressive language modeling paradigm as text-only LMs.
* Dynamic: The number of visual tokens is dynamically adjusted depending on the image content using a token selector and token merger. The token selector chooses the most crucial patches in the image, and the token merger reacquires the visual semantics of the unimportant patches, respectively.

The proposed method is pre-trained in two steps:
1. Tokenizer learning: The tokenizer is pre-trained to reconstruct the original image from the quantized compressed visual tokens.
2. Vision-language model learning: The VLM is pre-trained to predict the next visual/text token in a sequence, using the same autoregressive language modeling paradigm as text-only LMs.
The proposed method achieves high performance on multimodal understanding and generation tasks, with the relatively small size of model parameters and training data requirements. The paper also shows that quantization and token compression are effective techniques for improving the efficiency of VLM pre-training.

Overall, this paper presents a novel and effective method for VLM pretraining. The proposed method looks simple yet effective and achieves state-of-the-art results on various multimodal tasks.

**Strengths:**

The paper is well-organized and easy to read, and the advantages of the proposed tokenizer are clearly described and demonstrated through various experiments and evaluations. In particular, the method for improving the inefficiency of the conventional vision-text fusion method is effective, and this study can be considered primary research for further improvement.

**Weaknesses:**

There are a few suggestions for the ablation study.
1. Compare the performance of text generation using merged continuous embeddings and quantized visual tokens. The loss of detailed information due to quantization is likely to affect the performance of text generation. However, it would be interesting to see how much of a difference there is.
2. Show the performance difference with and without token merger. It would help to understand the importance of the token merger module.

**Questions:**

1. What is the computational (speed) improvement during inference using a dynamic token length? It may be important information because training and inference efficiency may vary.

2. There are a few things that could be improved in the manuscript. Please see the following suggestions.

- Figure 1 (a): `Adater-Style` -> `Adapter-Style`
- The first line in Section 3: `the Large language model` -> `the large language model`
- The last line in `Token Selector` of Section 3.1 : `the binary decision M` -> `the binary decision mask M`
- Equation 3: $||l_{2}(\hat{x_i})-l_{2}(c_{j})||$ -> $||\hat{x_i}-c_{j}||_2$
- The 2nd line in Section 4.4: `the token predictor` -> `the token selector`

---

> ### Author Response · Authors · 2023-11-17
> **Response to Reviewer K5nK**
>
> Dear reviewer, we sincerely appreciate for recognizing our work and providing valuable feedback. Below are our responses to the raised questions in your review.
>
> 1. *Compare the performance of text generation using merged continuous embeddings and quantized visual tokens. The loss of detailed information due to quantization is likely to affect the performance of text generation. However, it would be interesting to see how much of a difference there is.*
>
> * Yes, the loss of detailed information due to quantization will affect the multi-modal understanding performance. The comparisons between using merged continuous embeddings and quantized visual tokens have been presented in Table 7 (a) of the appendix. For your convenience, we have listed the detailed results in the following Table:
>
> | Setting |   Flickr | VQAv2 | OKVQA |
> | :-----:| :----:|:----: |:----: |
> | Quantized Visual Tokens | 70.3 |  53.0  | 44.8 |
> | Merged Continuous Embeddings | **72.7**   |  **56.4**   | **46.2** |
>
> * As observed, the continuous visual features achieve better multi-modal understanding performance than quantized visual codes. This phenomenon is reasonable as quantized embeddings represent the centroid of continuous visual features, potentially leading to the loss of visual details. Therefore, when visual input serves as the condition (i.e., sequence like [image, text]), we use the merged continuous visual features as the input to LLM.
>
> 2. *Show the performance difference with and without token merger. It would help to understand the importance of the token merger module.*
>
> * Thanks for your constructive suggestions. We have added the ablation study about the effect of the proposed token merger. From the presented results, it can be observed a distinct performance drop in multi-modal understanding tasks when eliminating the token merger module. Especially for visual question answering tasks that need fine-grained visual information as the reference, directly dropping the unselected visual tokens will lose some visual details. The proposed token merger can preserve the image details to some extent by compressing the unselected image patches onto the retained ones.
>
> | Setting  | Flickr | VQAv2 | OKVQA | GQA |
> | :-----:| :----:|:----: |:----: |:----: |
> | w/o Token Merger | 70.8 |  51.9  |  42.1 |  39.7  |
> | w/ Token Merger | **72.7**   |  **56.4**   | **46.2** | **42.3** |
>
>
> 3. *What is the computational (speed) improvement during inference using a dynamic token length? It may be important information because training and inference efficiency may vary.*
>
> * We compared the text-to-image generation speed of the proposed dynamic tokenizer and static tokenizer (fixed 256 sequence length) for 100 text prompts on a single NVIDIA A100 GPU with batchsize=1. The average time costs and average numbers of generated visual tokens per image are reported in the following Table. The proposed dynamic tokenizer generates 89 visual tokens on average for each text prompt, about 35% of the static one. Since the attention computation in LLM exhibits a quadratic relationship with respect to the token length, the sparsification of the dynamic tokenizer can thus accelerate the per-batch image synthesis speed by 120%.
>
> | Setting  | Average Visual Token  | Time per Image (s) |
> | :-----:| :----:|:----: |
> | Static Tokenizer | 256 |  9.93  |
> | Dynamic Tokenizer | 89  |  4.52  |
>
>
> 4. *There are a few things that could be improved in the manuscript. Please see the following suggestions*
>
> * Thanks for pointing out these typos in the draft. We have corrected them in the revision following your valuable suggestions.
>
> Thank you again for the detailed reviews of our paper and providing insightful suggestions. Please let us know if there are further questions or concerns.

---

### Official Review · Reviewer_6o4w · 2023-11-06

**Soundness:** 3 good
**Presentation:** 3 good
**Contribution:** 3 good
**Rating:** 6
**Confidence:** 4

**Summary:**

This work proposed to unify the pre-training objectives for visual and textual tokens while adapting the LLM for vision-language tasks. To this end, they first train a visual codebook to quantize the visual embeddings and assign each visual token a discrete label. To reduce the length of visual tokens, a token selector is proposed to select important visual tokens dynamically, and the token merger is used to merge dropped tokens with selected ones. The visual tokens and text tokens are then concatenated and fed to the LLM, which is trained for classification-based language modeling tasks. The authors conduct extensive experiments to evaluate the proposed methods, and the experimental results show that the proposed methods achieve state-of-the-art performance.

**Strengths:**

- The experimental results demonstrate that the proposed method achieves a significant improvement compared to previous works.
- The improvements from each proposed component is well supported by the ablation study results.

**Weaknesses:**

- The authors train a visual codebook to assign discrete labels to visual tokens, which can be computationally costly. In the context of efficiently adapting LLM for vision-language tasks, it would be interesting to explore whether existing codebooks [1, 2, 3] can be utilized to provide useful supervision for training LLM.

- Hard to be completely fair when compared against other baselines. For instance, compared to BLIP-v2, this work adds additional English text corpus and further fine-tunes the LLM. The significant performance improvement over previous work may be attributed to the additional training data and trainable parameters.

- The implementation details of the ablation studies could be clearer. It is confusing why the proposed method achieves different results in Tables 3, 6, and 7 when doing ablation studies.

- The authors claim that the learned codebook encodes high-level semantic concepts. However, the visualization results of codebooks in Figure 5 may suggest that the same code groups image patches based on low-level visual patterns, such as color and textures. It would be interesting to see additional image-level visualization results, similar to those in [4], to demonstrate that the codebook encodes high-level semantic concepts.

- Minor suggestions/typos: I noticed several typos while reading the paper:

  - Figure 1 (a) caption: Adater -> adapter
  - The symbols X and when calculating K and V of Equation (2) should be X_d.

---
- [1] Chen, Yuxiao, et al. "Revisiting multimodal representation in contrastive learning: from patch and token embeddings to finite discrete tokens." Proceedings of the IEEE/CVF Conference on Computer Vision and Pattern Recognition. 2023.
- [2] Esser, Patrick, Robin Rombach, and Bjorn Ommer. "Taming transformers for high-resolution image synthesis." Proceedings of the IEEE/CVF conference on computer vision and pattern recognition. 2021.
- [3] Bao, Hangbo, et al. "Beit: Bert pre-training of image transformers." arXiv preprint arXiv:2106.08254 (2021)
- [4] Peng, Zhiliang, et al. "Beit v2: Masked image modeling with vector-quantized visual tokenizers." arXiv preprint arXiv:2208.06366 (2022).

**Questions:**

Refer to the weakness.

---

> ### Author Response · Authors · 2023-11-17
> **Response to Reviewer 6o4w - Part1**
>
> We sincerely appreciate the time and effort the reviewer dedicated to reviewing our paper and providing constructive comments. Below are our responses to the raised concerns.
>
> 1. *The authors train a visual codebook to assign discrete labels to visual tokens ... it would be interesting to explore whether existing codebooks can be utilized to provide useful supervision for training LLM.*
>
> * We fully agree with the reviewer that exploring the influence of existing codebooks is meaningful. Actually, in our earlier experiments, we have tested the impact of different codebooks (e.g., VQ-VAE [1,2] and BEIT-2 [3]), but found that they were not effective in serving as a good visual tokenizer for multimodal LLM. The detailed comparisons between these codebooks and our proposed dynamic one are shown in the following Table. All these ablations are conducted on the same 30M pre-training data with clip ViTL/14 as the visual encoder.
>
> | Method | Codebook Size |  Flickr | GQA | VQAv2 | OKVQA |
> | :-----:| :----: |  :----: | :----:|:----: |:----: |
> | VQ-VAE | 8192 | 60.5 |  35.0  | 48.8 | 40.2 |
> | BEIT-2 | 8192 | 65.8  | 38.3 | 51.1 | 41.4 |
> | Ours | 8192 | **70.0** |  **40.8**  | **53.0** | **44.5** |
>
> * As observed, directly applying these existing codebook techniques to train LaVIT leads to sub-optimal downstream performance. We speculate that this may be due to the fact that the current codebooks were not explicitly tailored to accommodate text-oriented language models. For example, VQ-VAE's codebook is learned by reconstructing image pixels, which may cause the learned code to focus on low-level visual patterns and be incompatible with high-level word tokens. Although BEIT-2's codebook is derived by distilling high-level features from the CLIP teacher model, their visual codes do not have the 1D causal dependency and thus are not well compatible with LLM. Moreover, most of these existing codebooks tokenize an image into a long, fixed sequence (e.g., 1024 for DALLE [2]). Such a long sequence will invariably result in low efficiency, especially for generating images. Therefore, we introduce a well-designed dynamic visual tokenizer for better compatibility with LLM in the unified training.
>
> 2. *Hard to be completely fair when compared against other baselines ... The significant performance improvement over previous work may be attributed to the additional training data and trainable parameters.*
>
> * In terms of data, for a fair comparison, we employ the same multi-modal pre-training data (about 93M samples) as BLIP-2 [4] to exclude performance gains from additional data usage This data scale is smaller than that of most other methods. For example, the concurrent work Emu [5] uses 2.6B data, and KOSMOS [6] adopts about 3.1B image-caption pairs. Besides, the purpose of including English text corpus during pre-training is to preserve the already learned language understanding ability of LLM (e.g., the performance on linguistic benchmarks like MMLU) when acquiring good multimodal capabilities. The leveraged text corpus is Open-Source data (Redpajama [7]) for training the LLaMA model from scratch, which therefore, does not introduce new textual data and is just for alleviating the forgetting of previously learned knowledge.
>
> * While LaVIT fine-tunes the LLM compared to BLIP-2, our objective is to enable the visual modality to integrate better with LLM like a foreign language. In this way, text-oriented LLM can not only understand visual modality input but also generate visual content like language. In contrast, BLIP-2 can only comprehend the visual inputs but lacks the capability to generate them. Moreover, tuning the LLM instead of the adapter with limited trainable parameters can leverage the remarkable reasoning capabilities of LLM to learn the interaction across different modalities.
>
> 3. *The implementation details of the ablation studies could be clearer. It is confusing why the proposed method achieves different results in Tables 3, 6, and 7 when doing ablation studies.*
>
> * All the ablation studies were conducted on part of pre-training data (about 30M image-text pairs) by using the ViTL/14 of CLIP as the visual encoder. The language model is scheduled to train for 10K steps in these experiments. All other hyper-parameters are set to the same values as those listed in Table 9 of the appendix. Due to the costly training resources, these ablation studies did not progress to the final 10K steps and were stopped when the model performance did not change much. But it's important to note that we make sure that the same step was taken in each set of experiments for a fair comparison. For example, the experiments in Table 3 (a) are stopped at 9K steps, Table 6 for 9.5K steps and Table 7 (a) for 9.2K steps. As a result, the reviewer will find slight performance differences (within 1% change) in Tables 3, 6, and 7.

---

> ### Author Response · Authors · 2023-11-17
> **Response to Reviewer 6o4w - Part2**
>
> 4. *The authors claim that the learned codebook encodes high-level semantic concepts. However, the visualization results ... to demonstrate that the codebook encodes high-level semantic concepts.*
>
>  * Thanks for your constructive suggestions. We have updated the visualizations for the learned codebook in the draft and added more examples of image patches belonging to the same visual code. As observed in Figure 8 of the appendix, the learned discrete visual codes can convey explicit high-level visual semantics. For example, code 13014 represents the human arm, code 7260 shows the part of the car, code 15235 represents the bus, and code 15090 indicates the donuts. These visualizations demonstrate the interpretability of the learned codebook.
>
> 5. *Minor suggestions/typos: I noticed several typos while reading the paper*
>
>  * Thanks for pointing out these typos and we have corrected them in the draft.
>
> We hope these responses have settled your concerns. Thank you again for your detailed and valuable feedback. Please let us know if there are further questions or concerns.
>
> ---
>
> [1] Esser, Patrick, et al. Taming transformers for high-resolution image synthesis, CVPR 2021
>
> [2] Aditya Ramesh, et al. Zero-shot text-to-image generation, ICML 2021
>
> [3] Peng, Zhiliang, et al. Beit v2: Masked image modeling with vector-quantized visual tokenizers, arXiv preprint 2022
>
> [4] Junnan, et al. Blip-2: Bootstrapping language image pre-training with frozen image encoders and large language models, ICML 2023
>
> [5] Quan Sun, et al. Generative pretraining in multimodality, arXiv preprint 2023
>
> [6] Shaohan, et al. Language is not all you need: Aligning perception with language models, arXiv preprint 2023
>
> [7] Redpajama: An open source recipe to reproduce llama training dataset, https://github.com/togethercomputer/RedPajama-Data

---

### Author Response · Authors · 2023-11-17
**Response to All Reviewers**

We sincerely appreciate all the reviewers for their thoughtful and constructive feedback that helped us improve our submission. We have revised the manuscript and added clarifications based on the reviews. The detailed revision we made is summarized as follows:

* Re-correct the typos in writing mentioned by Reviewer 6o4w and K5nK.

* Add the ablation results and discussions in Appendix B.2 (Table 8) about the role of token merger.

* Add more visualization examples about the learned codebook in Appendix C (Figure 8) suggested by Reviewer 6o4w.

* Add the visualization for multi-modal in-context image generation in Appendix C (Figure 9) suggested by Reviewer 54hM.

* Modify the structure of the Method Section suggested by Reviewer 54hM.

* The discussion in Appendix E for differences in pixel decoder between ours and Emu mentioned by the Reviewer L5Te.

These changes have been highlighted in blue font.

---

### Meta-Review · Area_Chair_4N3r · 2023-12-05

**Metareview:**

This submission received three positive scores and one negative score. The novelty of the dynamic discrete tokens is conformed by reviewers. After reading the paper, the review comments and the rebuttal, the AC think the major concerns are about the presentation and experimental details, which are encouraged to be solved in the camera-ready paper.

**Justification For Why Not Higher Score:**

N/A

**Justification For Why Not Lower Score:**

N/A

---

### Decision · Program_Chairs · 2024-01-16

Accept (poster)